# Parts–per–million of ruthenium catalyze the selective chain–walking reaction of terminal alkenes

Sergio Sanz-Navarro[1,4], Marta Mon [1,4], Antonio Doménech-Carbó [2], Rossella Greco[1], Jorge Sánchez-Quesada [3], Estela Espinós-Ferri[3] & Antonio Leyva-Pérez [1✉]

The chain–walking of terminal alkenes (also called migration or isomerization reaction) is currently carried out in industry with unselective and relatively costly processes, to give mixtures of alkenes with significant amounts of oligomerized, branched and reduced by–products. Here, it is shown that part–per–million amounts of a variety of commercially available and in–house made ruthenium compounds, supported or not, transform into an extremely active catalyst for the regioselective migration of terminal alkenes to internal positions, with yields and selectivity up to >99% and without any solvent, ligand, additive or protecting atmosphere required, but only heating at temperatures >150 °C. The resulting internal alkene can be prepared in kilogram quantities, ready to be used in nine different organic reactions without any further treatment.

[1] Instituto de Tecnología Química (UPV–CSIC), Universidad Politècnica de València–Consejo Superior de Investigaciones Científicas, Avda. de los Naranjos s/n, 46022 Valencia, Spain. [2] Departament de Química Analítica, Universitat de Valencia, Dr Moliner, 50, 46100 Burjassot Valencia, Spain. [3] International Flavours & Fragrances Inc., Avda Felipe Klein 2, 12580 Benicarló, Castellón, Spain. [4]These authors contributed equally: Sergio Sanz-Navarro, Marta Mon. ✉email: anleyva@itq.upv.es

Alkenes are not only fundamental chemicals in the manufacturing chain, with millions of tonnes produced per year for the synthesis of polymers, detergents, lubricants, cosmetics and fragrances[1], but also the most prevalent functional group in natural products[2]. However, the synthesis of internal alkenes is still comparative expensive respect to terminal alkenes. Indeed, internal alkenes have a price generally between 5 and 50 times higher than the corresponding terminal alkenes in representative chemical suppliers, which clearly compromises the economic viability of subsequent transformations. Any synthetic method able to decrease the final price of the internal alkene, with high selectivity, will have a wide impact in the whole chemical manufacturing chain[3].

The most direct way to synthesize internal alkenes is the chain–walking reaction of terminal alkenes[4,5], and this transformation is used in petrochemistry with energy–intensive process (>250 °C). However, the resulting internal alkene is often accompanied of ramified, oligomerized and reduced by–products[6]. This mixture is not suitable for many applications, and in particular for the fine chemistry industry, which prefers to use the alkene metathesis or Wittig–type reactions, more selective but less atomically–efficient and waste–generating. With these data in mind, it is not surprising that a selective chain–walking reaction methodology, of application in a wide range of organic molecules, is still of high interest[7–9]. Figure 1a shows some representative examples[10–14] reported recently, which unfortunately employ metal loadings, additional ligands, additional additives or solvents which are far from being economically viable at industrial scale[15,16].

We and others have shown that part–per–million (ppm) amounts of different metals evolve to a cocktail of single atoms, clusters and ultra–small nanoparticles when dissolved and heated in an organic substance, regardless the initial state of the metal, and that some of the so–formed metal species can be catalytically active for a particular reaction[17–25]. Here, we show that tiny amounts of Ru species formed from a variety of Ru sources may catalyze, very efficiently, the alkene isomerization reaction.

## Results

**Catalytic tests with different Ru sources.** A 0.01 mol% (100 ppm) of different salts and complexes of alkenophilic metals were added to methyl eugenol **1** at 150 °C, in order to study the possible isomerization of the terminal alkene. The results are shown in Fig. 1b, and while Fe, Co, Ni and Cu did not show any catalytic activity (entries 1–8), according to gas chromatography coupled to mass spectrometry (GC–MS) and $^{1}$H– and $^{13}$C–nuclear magnetic resonance (NMR) analyses, Pd, Rh and Ir gave moderate conversions (entries 9–13), and Ru gave nearly quantitative conversions to methyl isoeugenol **2**, with a *trans:cis* ratio ~8:1 (entries 14–15). Figure 1c compares the cost for the production of one kilogram of **2** with some reported catalytic systems and the Ru(methylallyl)$_2$(COD) catalyst reported here (entry 15 in Fig. 1b), and it can be seen that the latter decreases the catalyst costs in at least two orders of magnitude.

Figure 2a shows that the isomerization of **1** with 10 ppm of Ru(methylallyl)$_2$(COD) (see Supplementary Fig. 1 for other catalytic amounts) follows an Arrhenius behavior and doubles the reaction rate each 10 °C temperature increase [$Q_{10}$ calculated = 1.9(8)], thus a relatively wide window of reaction temperatures can be applied to either decrease the reaction time or the Ru amount, or both. The Ru source was selected on the basis of its low price and bench stability (Supplementary Fig. 2 and 3) since Ru$_3$(CO)$_{12}$ must be handled under nitrogen[26]. A turnover frequency > $10^8$ h$^{-1}$ can be obtained (Supplementary Fig. 1). The reaction proceeded with the same result when an atmosphere of N$_2$ was placed instead of air. Figure 2b shows that RuO$_2$ is completely inactive and that RuCl$_3$ catalyzes the isomerization of **1** into **2** in moderate yield (entries 1–2) while, in contrast, not only Ru(methylallyl)$_2$(COD) and Ru(triphenylphosphine)$_2$Cl$_2$ catalyze

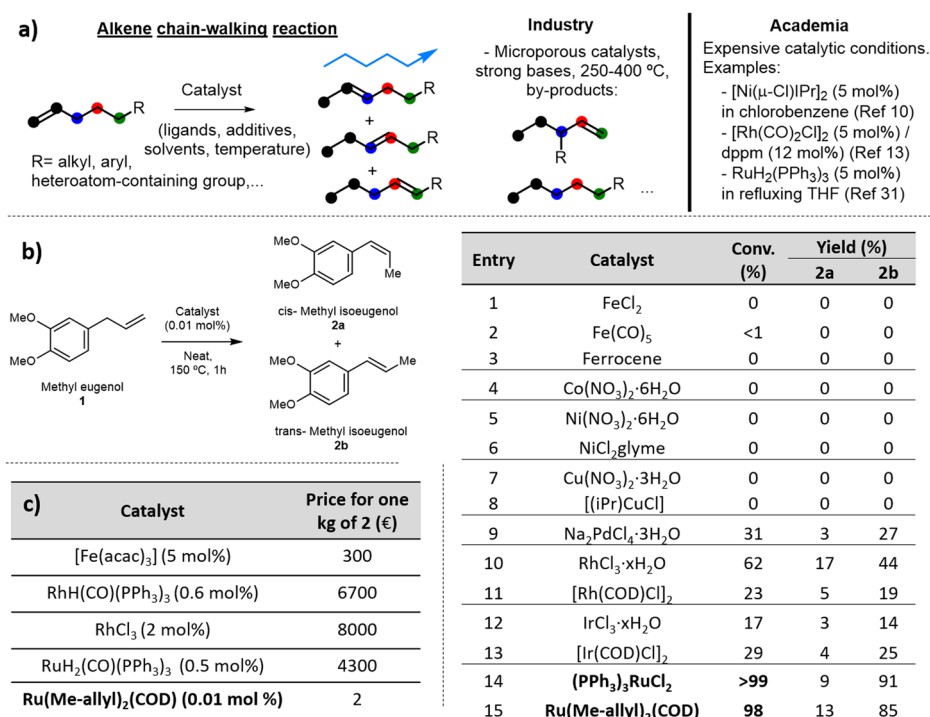

| Entry | Catalyst | Conv. (%) | Yield (%) 2a | Yield (%) 2b |
|---|---|---|---|---|
| 1 | FeCl$_2$ | 0 | 0 | 0 |
| 2 | Fe(CO)$_5$ | <1 | 0 | 0 |
| 3 | Ferrocene | 0 | 0 | 0 |
| 4 | Co(NO$_3$)$_2$·6H$_2$O | 0 | 0 | 0 |
| 5 | Ni(NO$_3$)$_2$·6H$_2$O | 0 | 0 | 0 |
| 6 | NiCl$_2$glyme | 0 | 0 | 0 |
| 7 | Cu(NO$_3$)$_2$·3H$_2$O | 0 | 0 | 0 |
| 8 | [(iPr)CuCl] | 0 | 0 | 0 |
| 9 | Na$_2$PdCl$_4$·3H$_2$O | 31 | 3 | 27 |
| 10 | RhCl$_3$·xH$_2$O | 62 | 17 | 44 |
| 11 | [Rh(COD)Cl]$_2$ | 23 | 5 | 19 |
| 12 | IrCl$_3$·xH$_2$O | 17 | 3 | 14 |
| 13 | [Ir(COD)Cl]$_2$ | 29 | 4 | 25 |
| 14 | (PPh$_3$)$_3$RuCl$_2$ | >99 | 9 | 91 |
| 15 | Ru(Me-allyl)$_2$(COD) | 98 | 13 | 85 |

| c) Catalyst | Price for one kg of 2 (€) |
|---|---|
| [Fe(acac)$_3$] (5 mol%) | 300 |
| RhH(CO)(PPh$_3$)$_3$ (0.6 mol%) | 6700 |
| RhCl$_3$ (2 mol%) | 8000 |
| RuH$_2$(CO)(PPh$_3$)$_3$ (0.5 mol%) | 4300 |
| **Ru(Me-allyl)$_2$(COD) (0.01 mol %)** | **2** |

**Fig. 1 Parts–per–million (ppm) of Ru catalyze the chain–walking reaction of terminal alkenes. a** The general alkene chain–walking reaction and some precedents and drawbacks in industry and academia. **b** Catalytic results for the isomerization of methyl eugenol **1** with 0.01 mol% (100 ppm) of different metal salts and complexes in the neat alkene at 150 °C for 1 h. **c** Price of catalyst required to get one kilogram of methyl isoeugenol **2** (**2a** + **2b**).

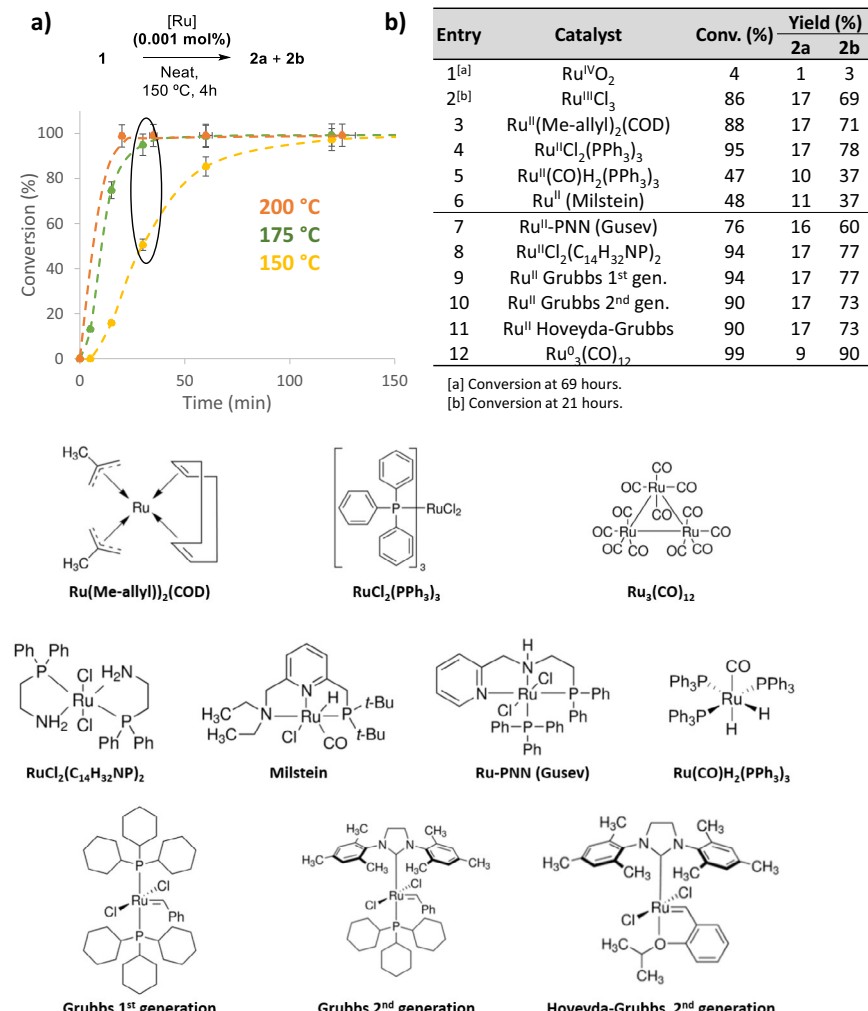

| Entry | Catalyst | Conv. (%) | Yield (%) | |
|---|---|---|---|---|
| | | | 2a | 2b |
| 1[a] | Ru$^{IV}$O$_2$ | 4 | 1 | 3 |
| 2[b] | Ru$^{III}$Cl$_3$ | 86 | 17 | 69 |
| 3 | Ru$^{II}$(Me-allyl)$_2$(COD) | 88 | 17 | 71 |
| 4 | Ru$^{II}$Cl$_2$(PPh$_3$)$_3$ | 95 | 17 | 78 |
| 5 | Ru$^{II}$(CO)H$_2$(PPh$_3$)$_3$ | 47 | 10 | 37 |
| 6 | Ru$^{II}$ (Milstein) | 48 | 11 | 37 |
| 7 | Ru$^{II}$-PNN (Gusev) | 76 | 16 | 60 |
| 8 | Ru$^{II}$Cl$_2$(C$_{14}$H$_{32}$NP)$_2$ | 94 | 17 | 77 |
| 9 | Ru$^{II}$ Grubbs 1$^{st}$ gen. | 94 | 17 | 77 |
| 10 | Ru$^{II}$ Grubbs 2$^{nd}$ gen. | 90 | 17 | 73 |
| 11 | Ru$^{II}$ Hoveyda-Grubbs | 90 | 17 | 73 |
| 12 | Ru$^0_3$(CO)$_{12}$ | 99 | 9 | 90 |

[a] Conversion at 69 hours.
[b] Conversion at 21 hours.

**Fig. 2 A variety of Ru catalysts and reaction temperatures are suitable for the chain–walking reaction of methyl eugenol 1 to methyl isoeugenol 2.** **a** Kinetics for the isomerization reaction of 1 g of **1** with Ru(methylallyl)$_2$(COD) at three different reaction temperatures; error bars account for a 5% uncertaintity. **b** Isomerization reaction catalyzed by 0.001 mol% (10 ppm) of different commercially available Ru compounds, the Ru oxidation state for each compound is indicated. For entries 1–6, the reaction was carried out with 4–16 g of **1** adding the solid catalyst directly. For entries 7–12, the reaction was carried out with 0.3–1 g of **1** adding a solution of the catalyst in dichloromethane. Below, structures of the different Ru complexes employed as initial Ru source.

the reaction at 0.001 mol% (10 ppm), but also typical Ru(II) hydrogenation (entries 5–8) and metathesis (entries 9–11) catalysts, and also a Ru(0) complex (entry 12), to give product **2** in good yields and with similar selectivity (for catalyst structures see Fig. 2). The observation of a 5–20 min induction time during the reaction profile of different Ru complexes (Supplementary Fig. 4) suggests that a common low–valent Ru species is formed in hot **1**, to catalyze the isomerization to **2**. Despite the activity of Ru for the isomerization of alkenes is known, this extreme catalytic activity of a variety of Ru species under solventless reaction conditions is, to our knowledge, unprecedented and remarkable[27].

**Reaction scope**. Figure 3 shows that a variety of terminal alkenes isomerizes to the corresponding internal alkenes when ultralow amounts of Ru(methylallyl)$_2$(COD) (down to 0.001 mol%, 10 ppm, i.e. product **2**) are used as a catalyst, including phenyl-propenes with completely different charge distribution in the aromatic ring (products **3–9**)[27], alcohols (products **10–13**) and longer carbon lineal chains containing other functional groups or not (products **14–25**, for starting alkene structures see Supplementary Fig. 5). The starting alkenes have not been previously

purified, and they come from very different feedstocks. The final mixture is the statistical composition according to a Boltzmann distribution, i.e. the thermodynamic products, and the turnover number (TON) can be as high as ~3·10$^6$ (i.e. products **21** and **25**, notice that several alkene migrations occur in the same molecule). The double bond migration occurs exclusively in methylene –CH$_2$– atoms, and alcohols transform into carbonyl groups when reached (products **10–13**)[28–31], since the chain–walking only stops when a further substitution is found. This regioselectivity allows to predict the reactivity of different alkenes and, indeed, geminal and internal alkenes are completely unreactive under these reaction conditions (Supplementary Fig. 6). With this regioselectivity in mind, the high–scale synthesis of 5–ethylidene–2–propoxycyclohexan–1–ol **24**, a commercial fragrance (Veraspice™) with an annual production >100 tonnes, could be performed. The current industrial production of **24** (Supplementary Fig. 7) requires the selective isomerization of the terminal alkene catalyzed by RhCl$_3$ (5000 ppm) in isopropanol solvent, which accounts for ~70% of the total material cost. Here, the Ru–catalyzed process allows to obtain kilograms of **24** with just 0.05-0.01 mol% (500–100 ppm) – of Ru, which translates in

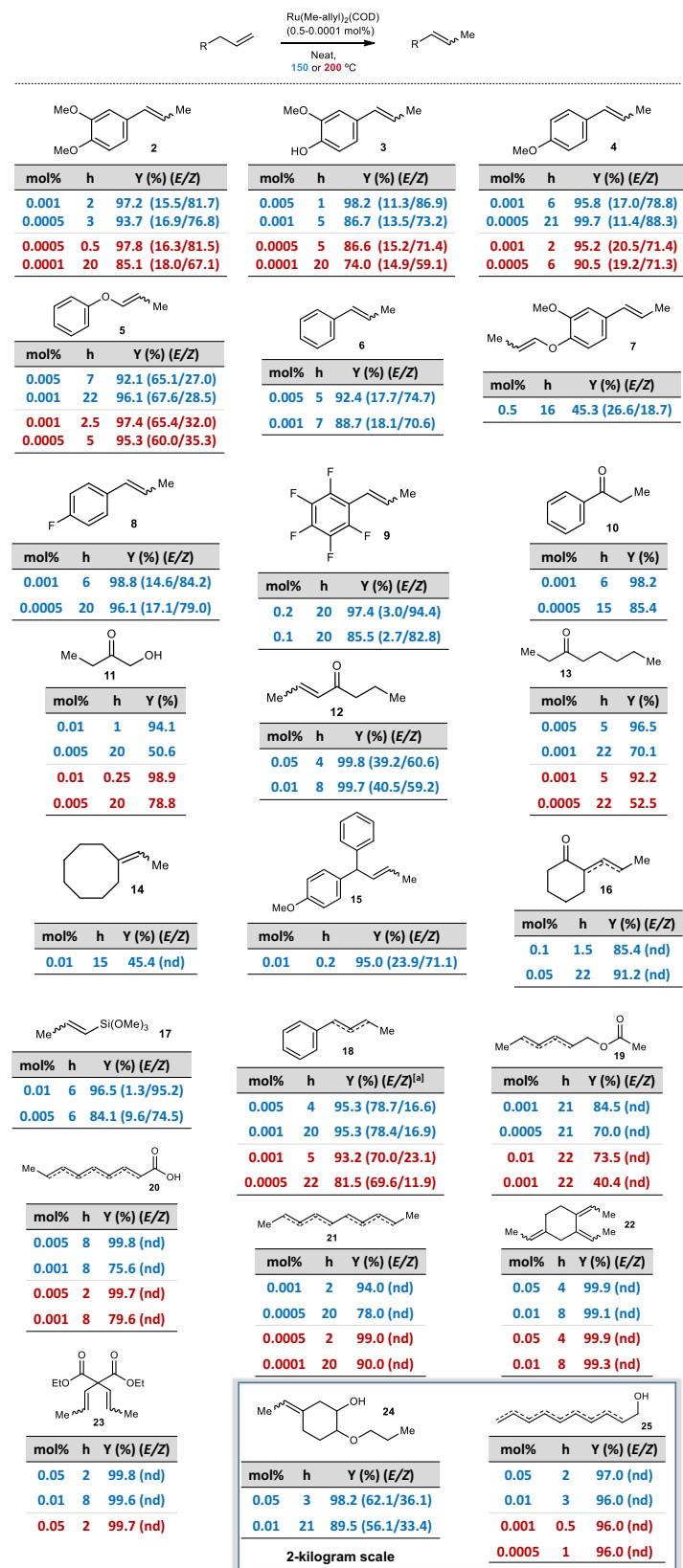

**Fig. 3 Scope of products from neat isomerization reactions with catalytic Ru(methylallyl)₂(COD).** The reactions are carried out at 5 gram scale, and also at 2 kg scale for the industrial fragrance products. The structure of the internal alkene products together with different reaction conditions are shown. Isolated yields, "nd" stands for not determined. [a] *E/Z* refers to 2–alkene/3–alkene.

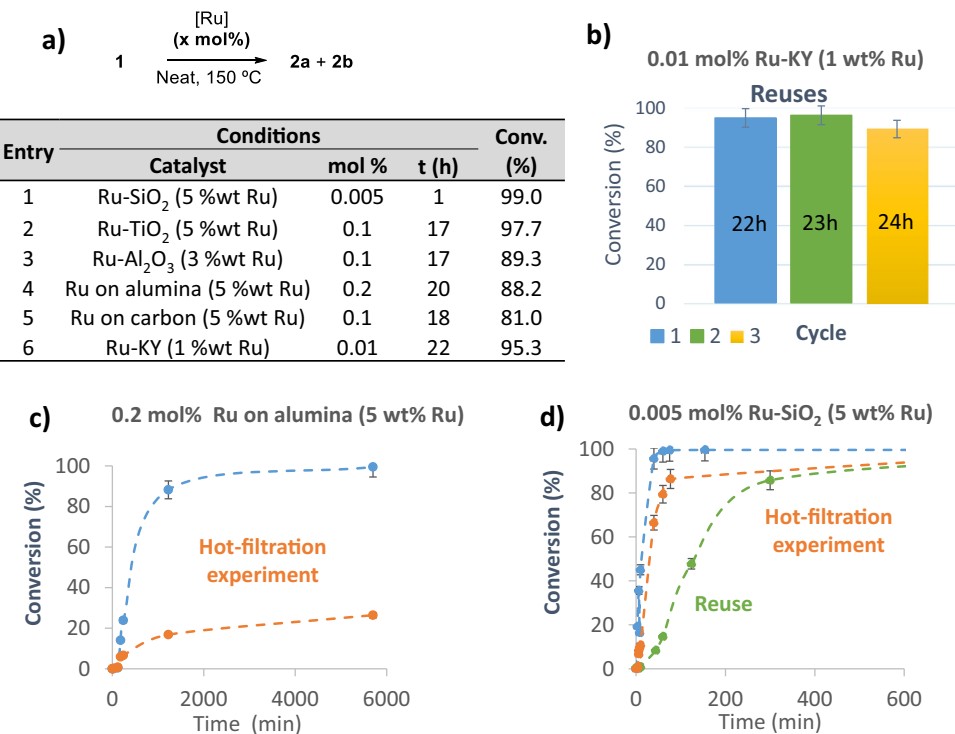

**Fig. 4 Supported Ru catalysts, reuse and leaching tests. a** Results for the isomerization of **1** with different Ru–supported solid catalysts. Ru–KY refers to zeolite NaY exchanged with K⁺ and then with Ru³⁺. The Ru wt% on each solid is indicated between brackets, mol% refers to the catalytic amount of Ru in reaction. **b** Reuses of Ru–KY. **c** Hot filtration leaching test for commercial Ru–Al₂O₃. **d** Hot filtration leaching test and reuse of Ru–SiO₂. The supported Ru catalysts are either commercial (entry 4 and 5) or in–house prepared by impregnation of aqueous RuCl₃ (entry 1–3 and 6). The optimized amount of catalyst is shown for each solid. Error bars account for a 5% uncertainty.

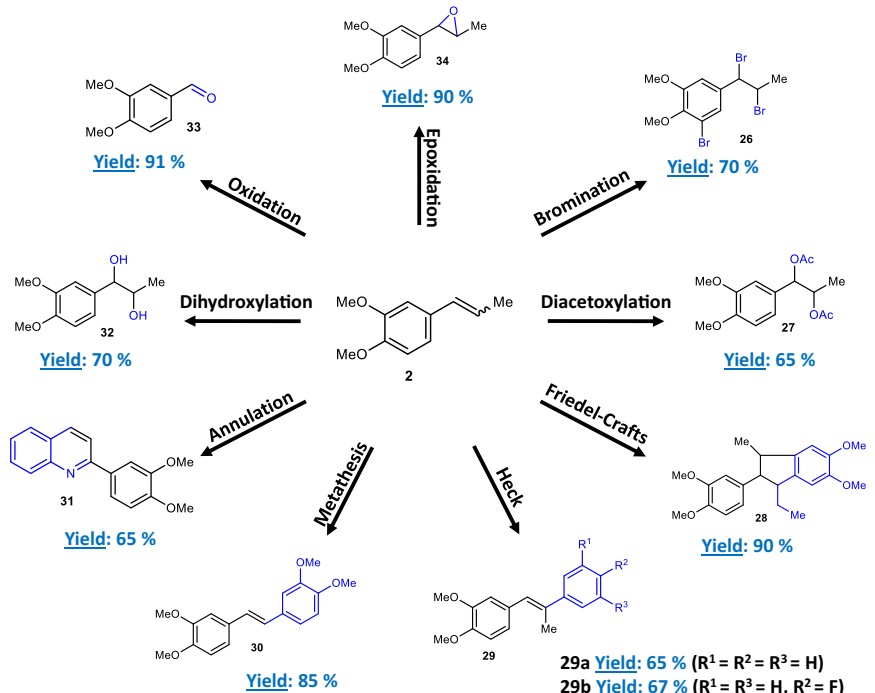

**Fig. 5 One-pot reactions of the internal alkene product methyl isoeugenol 2.** Representative reactions for internal alkenes were carried out in the same flask after the isomerization of **1** catalyzed by 10 ppm of Ru(methylallyl)₂(COD). Isolated yields.

<40% material cost. The final composition of the commercial fragrance, according to GC and NMR measurements, perfectly matches that obtained with $RhCl_3$, thus the olfative properties are not modified (Supplementary Fig. 8 and Supplementary Table 1). Attempts to optimize $RhCl_3$ in the same way that Ru(methy-lallyl)$_2$(COD) were unsuccessful since the absence of the iso-propanol solvent decreases the reaction rate (Supplementary Fig. 9). Following this result, the isomerization of the precursor for another industrial fragrance, i.e. 9–decen–1–ol (Rosalva™) to Isorosalva™ **25**, was attempted, and 5 ppm of Ru(methy-lallyl)$_2$(COD) at 200 °C were enough to quantitatively obtain the desired product. This process is industrially also performed with $RhCl_3$ (0.1 mol%), and the mixture of internal alkenes obtained here is identical to that obtained with $RhCl_3$, thus fragrance composition is maintained with a dramatic decrease in the pro-duction costs for **25** (Supplementary Table 2).

**Solid Ru catalysts: recovery and reuse.** Such an extremely low amount of Ru for the isomerization of terminal alkenes in batch seems attractive enough for industrial use, however, the use of solid catalysts is generally preferred from a practical point of view. Figure 4a shows the catalytic results for the isomerization of **1** with Ru–supported solids. It can be seen that all the supports tested here catalyze well the isomerization, with Ru amounts ranging between 50 and 1000 ppm under solventless conditions. Ru can be supported in cationic form, reduced under $H_2$ or not, or alternatively, commercially available Ru nanoparticles on alumina or carbon can be used. Ru–KY refers to zeolite NaY exchanged with $K^+$ and then with $Ru^{3+}$, and Fig. 4b shows that this catalytic solid is reusable, and the hot filtration test in Fig. 4c indicates that the catalytic activity from species in solution in Ru–alumina is less, although appreciable. In accordance, Fig. 4d shows that a decrease in the initial rate from use to use of the solid catalyst Ru–SiO$_2$ is observed. However, the Ru amount leached into the solution is below the detection limit of inductively–coupled plasma-atomic emission spectroscopy (ICP–AES) measurements (<0.1 ppm), after cooling the mixture, and the solid catalyst can be reused with nearly the same final yield of product **2** after 24 h. These results can be explained by Ru redeposition on the support at room temperature[32], and bring the possibility of employing reusable catalysts for this isomerization reaction[33,34].

**One-pot reactions.** Figure 5 shows that the Ru–catalyzed iso-merization reaction can be easily engaged in one-pot with a variety of reactions, since the internal alkene is obtained neat with just a few ppm of Ru isomerization catalyst remaining, thus any further treatment (solvent evaporation, filtration, washings…) is not required[35,36]. Not only classical oxy–addition reactions to the double bond such as the epoxidation, dihydroxylation, alkox-ycarbonylation, oxidative breaking and diacetoxylation reactions of **2** can be carried out[37], but also metal–catalyzed carbon–carbon bond forming reactions such as alkene metathesis[38] and Mizoroki–Heck reaction[25] can be directly performed on the same flask after the isomerization of **1**. These results illustrate the simplicity of the solventless Ru–catalyzed isomerization proce-dure shown here, telescopable in one-pot during synthetic routes[39,40].

**Reaction mechanism.** The mechanism of the isomerization reac-tion was studied through kinetic, spectroscopic, voltammetry, iso-topic and scrambling experiments, and also by computational methods. The rate equation obtained for the isomerization of **1–2** is $v_0 = k_{exp}[\mathbf{1}][Ru]$ (Supplementary Fig. 10), and the addition of HCl (gas) or 1,8–diazabicyclo[5.4.0]undec–7–ene (DBU) hampers the

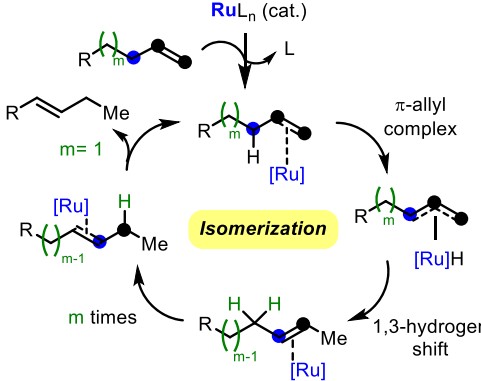

**Fig. 6 Proposed reaction mechanism catalyzed by Ru (possibly Ru$^{II}$).** The isomerization occurs $m + 1$ times along the chain until a more substituted position is found. Ru acts as a shuttle for the H atom and the alkene to move along the chain.

reaction (Supplementary Fig. 11). These results suggest that the Ru active species are single $Ru^{n+}$ atoms, in accordance with the fact that $Ru^{II}$ and not (agglomerated) $Ru^0$ is the plausible active oxidation state as assessed by Fourier–transformed infrared spectroscopy (FTIR, Supplementary Fig. 12)[41], flow injection–mass spectrometry with Orbitrap analyzer (HPLC–Orbitrap MS, Supplementary Fig. 13)[42], cyclic voltammetric (CV, Supplementary Fig. 14)[43–45] and FTIR with CO as a probe molecule on Ru–supported solids (Sup-plementary Fig. 15)[46,47] measurements. Kinetic experiments with deuterated alkene **allylbenzene-$d2$–$2$** show some involvement of the C–H breaking during the rate–determining step (rds), since a KIE = 1.4(5) is obtained (Supplementary Fig. 16). Ru–hydride complexes do not catalyze better the isomerization of **1** than a plethora of non–hydride complexes (see Fig. 2 above), and an experiment where **allylbenzene-$d2$ –$2$** was mixed with **1** does not show any scrambling of hydrogen atoms between the alkenes (Supplementary Fig. 17). The use of $CCl_4$ or (2,2,6,6-tetra-methylpiperidin-1-yl)oxyl (TEMPO, as an additional radical quencher) did not show any change in the reaction rate, however, in contrast, ethylenediaminetetraacetic acid did produce a significant decrease in the isomerization rate of **4**, which was used instead of **1** to test higher amounts of Ru catalyst and additives (Supplementary Fig. 18). These results point to a cationic intermediate Ru complex. In accordance, density functional theory (DFT) calculations, for both $Ru^{II}$ and $Ru^I$ oxidation states, show that the chain walking of the Ru atom in a model terminal alkene (1-butene, this alkene is also reactive under the conditions in Fig. 4a, using Ru/C as a catalyst) is energetically favored in, at least, 13 kcal mol$^{-1}$, that $Ru^{II}$ stabilizes twice the π–allyl intermediate than $Ru^I$, and that a *per*alkene $Ru^{II}$ complex suffers a smooth complex formation and 1,3–hydrogen shift steps (not higher than 4 kcal mol$^{-1}$) and then a barrierless isomerization (Supplementary Fig. 19 and Table 3). All these results, together, point to a π–allyl (1,3–hydrogen shift) rather than a σ–alkyl (1,2–hydrogen shift) mechanism (Supplementary Fig. 20)[48]. Figure 6 shows the plausible mechanism for the Ru–catalyzed alkene iso-merization reaction reported here. First, the tiny amounts of starting Ru compound transform into a catalytically active or just in resting state *per*–alkene $Ru^{II}$ complex when heated in the neat alkene, internally walking along the alkene by a classical π–allyl mechanism[49,50] until the catalyst finds a substituted carbon atom where the catalyst definitively stops, and starts the catalytic cycle with a new terminal alkene molecule. It must be notice here that, at this point, it is difficult to exactly know the structure of the active complex, however, according to previous results, a fully coordinated, presumably square planar 16–electron *per*–alkene $Ru^{II}$ complex is plausible a resting state while a more reactive 14–electron *tris*–alkene

Ru[II] would be more favorable for catalysis[51]. A Ru[II](olefin)$_x$H$_2$ and its possible tetramer, {Ru[II](olefin)$_x$H$_2$}$_2$, must not be discarded as a potential catalyst resting state or actual catalyst on the basis of literature precedents with related {HIr(olefin)2}$_4$ complexes[52,53]. The fact that not only the catalytic activity but also the selectivity for a plethora of different Ru complexes is extraordinarily similar (see Fig. 2 above), and that if one considers that the formation of three new Ru[II]-olefin bonds of around 25–35 kcal mol$^{-1}$ plus one Ru-H bond of ~70 kcal mol$^{-1}$,[54] with its *trans* effect, might be enough to labilise even the strong ligand bonds, strongly supports our original hypothesis that a common Ru species is formed under solvent–free heating conditions[18,55]. Indeed, in-situ $^{31}$P NMR experiments, in solution, showed how the PPh$_3$ ligands of the stable complex Ru(PPh$_3$)$_3$Cl$_2$ complex come off under reaction conditions (Supplementary Fig. 21).

In conclusion, a variety of terminal alkenes transform into the corresponding terminal alkenes when heated at >150 °C in the presence of part-per million amounts of practically any available ruthenium compound, including ruthenium supported on solids. Turnover frequencies of $10^8$ h$^{-1}$ are observed. To our knowledge, the amount of metal catalyst employed here is typically three order of magnitude lower than any other reported method where no solvent, additive, ligand or special atmosphere is required. This extremely simple technology gives access to internal alkenes at virtually the same price than terminal alkenes, to obtain, regioselectively, non–branched internal alkenes by circumventing the industrial proton–catalyzed process.

## Methods

**General**. Glassware was dried in an oven at 175 °C before use. Reactions were performed in 2.0 ml vials equipped with a magnetic stirrer and closed with a steel cap having a rubber septum part to sample out. Reagents and solvents were obtained from commercial sources and were used without further purification otherwise indicated, including the starting alkenes. Products were characterized by GC–MS, $^1$H– and $^{13}$C–NMR, and DEPT, and compared with the given literature. Gas chromatographic analyses were performed in an instrument equipped with a 25 m capillary column of 5% phenylmethylsilicone. *N*-dodecane was used as an external standard. GC/MS analyses were performed on a spectrometer equipped with the same column as the GC and operated under the same conditions. $^1$H, $^{13}$C and DEPT measurements were recorded in a 300 or 400 MHz instrument using CDCl$_3$ or DMSO as a solvent, containing TMS as an internal standard. The metal content of the solids was determined by inductively coupled plasma-atomic emission spectroscopy (ICP–AES) after disaggregation of the solid in isopropanol/water mixtures. Absorption spectra were recorded on a 300 UV–Vis spectrophotometer.

**Cyclic voltammetry**. Electrochemical experiments have been performed in 100 ppm solutions of the Ru complexes in neat alkene after adding an equal volume of 0.10 M Hex$_4$NPF$_6$/MeCN acting as an electrolyte. No deaeration was performed in order to reproduce the experimental conditions of catalytic experiments. Measurements were carried out at 298 ± 1 K. A conventional three-electrode electrochemical cell was used with a Pt wire pseudo-reference electrode, glassy carbon working electrode (GCE, BAS MF 2012, geometrical area 0.071 cm$^2$), and a platinum mesh auxiliary electrode. The potentials were calibrated relative to the ferrocenium/ferrocene (Fc$^+$/Fc) couple after addition of ferrocene until 0.5 mM concentration to the problem solutions. Cyclic and square wave voltammetry were used as detection modes.

**Orbitrap measurements**. The flow injection-HRMS consisted of an injection and pump systems and a single mass spectrometer Orbitrap Thermo Fisher Scientific (Exactive™) using an electrospray interface (ESI) (HESI-II, Thermo Fisher Scientific) in positive or negative mode. The injector was directly connected to the source and 10 μL of the sample was injected into the flow-injection solvent consisting of an aqueous solution of 0.1% formic acid and methanol (1:1). The flow rate remained at 0.20 ml min$^{-1}$ over 5 min. The ESI parameters were as follows: spray voltage, 4 kV; sheath gas (N$_2$, > 95%), 35 (non-dimensional); auxiliary gas (N$_2$, > 95%), 10 (non-dimensional); skimmer voltage, 18 V; capillary voltage, 35 V; tube lens voltage, 95 V; heater temperature, 305 °C; capillary temperature, 300 °C. The mass spectra were acquired employing two alternating acquisition functions: (1) full MS, ESI + , without fragmentation (higher collisional dissociation (HCD) collision cell was switched off), mass resolving power = 25,000 FWHM (full width at half-maximum); scan time = 0.25 s; (2) all-ion fragmentation (AIF), ESI + , with fragmentation (HCD on, collision energy = 30 eV), and mass resolving power =

10,000 FWHM; scan time = 0.10 s. The mass range was 150.0–1500.0 *m/z*. The chromatograms were processed using Xcalibur™ version 2.2, with Qualbrowser (Thermo Fisher Scientific).

**FTIR spectroscopy of adsorbed CO**. Fourier transform infrared (FTIR) spectra were recorded on a Biorad FTS–40 A spectrometer equipped with a DTGS detector, using CO as a probe molecule. The experiments have been carried out in a homemade IR cell able to work in the high and low (77 K) temperature range. Prior to CO adsorption experiments, the sample was evacuated at 423 K under vacuum (10$^{-6}$ mbar) for 1 h. CO adsorption experiments were performed at 77 K in the 0.2–20 mbar range. Spectra were recorded once complete coverage of CO at the specified CO partial pressure was achieved. Deconvolution of the IR spectra has been performed in the Origin software using Gaussian curves where the full width at half–maximum (fwhm) of the individual bands has been taken as constant. The peak areas are normalized to the sample weight. The in-situ reaction with 1-pentene was performed by bubbling a solution of 1-pentene in *n*-pentane (1:10 *v:v*) with Ar and passing the corresponding flow through the IR chamber containing the solid catalyst, during 10 min at 150 °C. After this, the IR chamber was evacuated and treated with CO as above.

**Computational**. Quantum chemistry calculations are based on DFT and were performed using the B3LYP functional in combination with the 6–311 G** and LANL2DZ basis sets, the first for C, H and O atoms and the latter for Ru, respectively. Harmonic vibrational frequencies were evaluated at the same level of theory in order to characterize true minima on the potential energy surface. All the calculations were performed using the Gaussian09 program.

**Reaction procedures**. Isomerization procedure of methyl eugenol **1** with different metal salts and complexes. Methyl eugenol **1** (1 g, 5.6 mmol) was charged in a 2 mL vial equipped with a magnetic stirrer and the corresponding catalyst (0.01 mol%) was added, dissolved in dichloromethane or methanol (5–10 μl). The vial was closed with a cap, placed in a steel block at 150 °C under magnetically stirring and maintained during the reaction time. Aliquots of the reaction mixture were taken to follow the reaction over time by GC and NMR. It should be noted that stock solutions were prepared to add the catalyst, since the amounts used are too small to be weighed. To prepare the solutions, volumetric flasks were used with dichloromethane or methanol as a solvent.

Isomerization procedure of methyl eugenol **1** with different Ru compounds. Methyl eugenol **1** (0.3–16 g) was charged in a vial or flask equipped with a magnetic stirrer and the corresponding catalyst (0.001 mol%) was added directly or dissolved in dichloromethane. The vial or flask was closed with a cap, placed in a pre–heated bath oil at 150 °C under magnetically stirring for a given reaction time. Aliquots of the reaction mixture were taken to follow the reaction over time by GC and NMR.

Kinetics for the isomerization reaction of methyl eugenol **1**. Methyl eugenol **1** (1 g, 5.6 mmol) was charged in a 2 mL vial equipped with a magnetic stirrer and a dichloromethane solution of Ru(methylallyl)$_2$(COD) (0.001, 0.0005 or 0.0001 mol%) was added. A stock solution of the catalyst in dichloromethane was prepared using volumetric flask. The vial was closed with a cap, placed in a steel block at the corresponding reaction temperature (150, 175 or 200 °C) under magnetically stirring and maintained during the reaction time. Aliquots of the reaction mixture were taken to follow the reaction over time by GC and NMR.

**General alkene isomerization procedure**. The terminal alkene (0.3–1 g) was charged in a 2 mL vial equipped with a magnetic stirrer and the Ru(methylallyl)$_2$(COD) catalyst (0.5–0.0001 mol%) was added, dissolved in dichloromethane. The vial was closed with a cap, placed in a steel block at the corresponding reaction temperature (150 or 200 °C) under magnetically stirring and maintained during the reaction time. Aliquots of the reaction mixture were taken to follow the reaction over time by GC and NMR.

**Preparation of solid catalysts**. The corresponding amount of the commercial support (SiO$_2$, TiO$_2$, Al$_2$O$_3$ or zeolite) were added to an aqueous solution of RuCl$_3$·xH$_2$O and magnetically stirred for 24 h at room temperature. After that time, the impregnated supports were filtered under vacuum suction and dried in an oven at 100 °C.

Supported Ru catalysts, reuse and leaching tests:
Isomerization procedure of methyl eugenol **1** with different supported Ru catalysts. Methyl eugenol **1** (0.5–1 g) was charged in a vial or flask equipped with a magnetic stirrer and the corresponding solid catalyst (0.2–0.005 Ru mol%) was added directly. The vial was closed with a cap, placed in a pre–heated bath oil at 150 °C under magnetically stirring for a given reaction time. Aliquots of the reaction mixture were taken to follow the reaction over time by GC and NMR.

**Reuses**. The general reaction procedure above was followed for different supported Ru catalysts. After 24 reaction time, the solid catalyst was recovered by filtration and washed with toluene. After drying, the catalyst was weighted and the reagent added in proportional amount to keep the initial relative molar ratios.

**Hot-filtration experiments**. Following the general reaction procedure above, the hot reaction mixture was filtered through a 0.25 μm Teflon filter into a new vial equipped with a magnetic stirrer and placed at the reaction temperature, and the filtrates were periodically analyzed by GC to compare with the results obtained with the solid catalyst still in.

Procedures of one-pot reactions from methyl isoeugenol **2**.

**Bromination**. Following the general procedure described above for the isomerization reaction, a mixture of neat **2** (89 mg, 0.50 mmol) and Ru(methylallyl)$_2$(COD) (0.8 μg, 0.0005 mol%) was heated in a 8 ml vial equipped with a magnetic stir bar at 200 °C. After cooling, CHCl$_3$ (2 ml, 0.25 M) was added, then bromine (176 mg, 1.1 mmol) dissolved in CHCl$_3$ (0.6 mL) was slowly added and the mixture was stirred at 25 °C for 2 h. Water was added and the phases separated. The organic phase was successively washed with an aqueous solution of 10% Na$_2$SO$_3$ and 1 M Na$_2$CO$_3$. The combined organic phases were dried over MgSO$_4$, filtered and concentrated under vacuum. The crude product was purified by chromatography on silica gel (20% AcOEt in hexane) to give 145 mg (70%) of **26** as a yellow oil.

The rest of procedures for one-pot reactions from **2** can be found in Supplementary Information.

Synthesis of deuterated alkene **allylbenzene-$d^2$** -.

SOCl$_2$ (0.8 ml, 10.8 mmol) and pyridine (35.5 μl, 0.45 mmol) were added to a solution of benzyl alcohol-$d^2$ (1 g, 9.0 mmol) in diethyl ether (9 ml, 1 M), and the mixture was stirred at 40 °C for 1 h. After that time, SOCl$_2$ (0.8 ml, 10.8 mmol) and pyridine (35.5 μl, 0.45 mmol) were added again and the mixture was stirred for 1 h at the same temperature. After cooling, water was added and extracted with diethyl ether and washed with brine. The combined organic phases were dried over MgSO$_4$, filtered and concentrated under vacuum. The solvent was evaporated under reduced pressure to give a crude product. Tributyl(vinyl)tin (1.4 ml, 4.7 mmol) and Pd(PPh$_3$)$_4$ (225 mg, 0.2 mmol) were then added to this crude, containing benzyl chloride-$d^2$ (500 mg, 3.9 mmol) in THF (13 ml, 0.3 M), and the mixture was stirred at 70 °C for 2 h. Then, the solvent was evaporated under reduced pressure to give the final crude product. Flash chromatography (1% AcOEt in hexane) gives 280 mg (60%) of **allylbenzene-$d^2$** - as a yellow oil.

Kinetic experiments with deuterated alkene **allylbenzene-$d^2$** -.

Following the general procedure described above for the isomerization reaction, a mixture of neat **allylbenzene-$d^2$** - (200 mg, 1.7 mmol) and Ru(methylallyl)$_2$(COD) (0.03 mg, 0.005 mol%) was heated in a 2 ml vial equipped with a magnetic stir bar at 150 °C. Aliquots of 7.5 μl were periodically taken and were diluted with ethyl acetate to analyze the reaction evolution by GC and GC–MS. The KIE value was obtained with the formula k$_H$/k$_D$, where the $k$ values are the initial rate of the reactions obtained by linear regression of the initial points in the linear part of the curve (0–10 min for **allylbenzene** , 5–20 min for **allylbenzene-$d^2$** -).

## Data availability

The datasets generated during and/or analyzed during the current study are included in this published paper (and its supplementary information files) or available from the corresponding author on request. Datasets could be also deposited in public repositories of the UPV and CSIC. Source data are provided with this paper.

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

## Acknowledgements

A.L.-P. thanks the financial support by IFF and MICIIN (PID2020–115100GB–I00). We also thank the funding for open access charge to the Universitat Politècnica de València. S.S.-N. thanks a fellowship from MINECO (project number CTQ 2017–86735–P). M.–Mon thanks MICIIN from a contract under the Juan de la Cierva program (FJC2019–040523–I). R.G. thanks a contract from the ITQ (SEV–2016–0683). We thank Dr. I. Domínguez for performing Orbitrap experiments. We thank V. Carbonell Vanaclocha for his help in the laboratory.

## Author contributions

M.M. and S.S.-N. performed and interpreted the experimental part. A.D.-C. performed and interpreted the electrochemical experiments. R.G. performed the computational part. J.S.-Q. and E.E.-F. designed the catalytic system for fragrance compounds, and supervised the experiments at high scale. A.L.-P. designed the experiments and supervised the whole work. The paper has been written with contributions from all authors.

## Competing interests

Patent EP21382234 has been presented to protect the synthesis of fragrance compounds by the methodology here reported, i.e., compounds **24** and **25**, and patent ES16411667 to protect unsubstituted long–linear carbon chain alkenes. S.S.-N., M.M., and A.L.-P. appear in the first patent, and M.M. and A.L.-P. appear in the second patent. S.S.-N., M.M., and A.L.-P. declare no other competing interests. The rest of authors declare no competing interests.
