## [Peer Review File · Nature Communications]

REVIEWER COMMENTS

Reviewer #1 (Remarks to the Author):

The authors make a convincing case for a more cost-effective transition metal alkene isomerization catalyst, and they provide a nice example. The Ru(methallyl)(COD) complex is devoid of higher molecular weight phosphines or strongly-held CO ligands that may in fact inhibit the catalysis. The results shown are highly significant and definitely deserving of publication after some revision, but the revisions are absolutely needed before publication. TON of 1×10^8 per h is quite amazing! But of course this is at 200 C. The use of supported materials and thorough characterization is notable. Also significant is that the very low amounts of ruthenium allow for subsequent catalyzed transformations.

There are several downsides however. Higher temperatures such as 150 or 200 C do lead to a loss of E/Z selectivity because there is no kinetic control, but if that loss is tolerable in the application then it does not matter. A number of the Figures need some work, some key literature is not cited, but as far as I can see, with the exception of some electrochemical work, only editorial changes would be needed to make the paper suitable for publication, but as the paper is now it is not acceptable.

It is very interesting that the system does not affect internal double bonds (Fig S6B) because it is so active with terminal ones. Can the authors comment? If a tris or tetrakis (terminal alkene) complex is the active catalyst, it could be quite bulky thanks to the presence of several alkenes. What about some (admittedly somewhat speculative) calculations in support of pi allyl hydride mechanism on Ru(II)(alkene)_n species?

What is the purity of the feedstocks? There are no details mentioned, unless I am missing something. I ask because over time, alkenes with allylic positions suffer some oxidation. If the feedstocks are not purified in any way, this would be a plus for the manuscript.

There are no references to significant work on alkene chain walking catalysis from the group of Sigman – e.g. JACS 2020, 142, 10516; J Org Chem 2014, 79, 11841; and beautiful work in Nature 2014, 508, 340.

The authors have not cited any of the considerable work by Grotjahn on alkene isomerization, including reactions that finish in less than 1 h at ambient temperature using only 500 ppm of catalyst, and taking 24 h with 100 ppm – I had to refresh my memory on this, but see in particular

JACS 2012 134, 10357. Clearly the Grotjahn catalyst is more expensive, but the fact that it operates at ambient temperatures is useful – it even works below zero deg C! – see ACS Catal. 2020, 10, 15250. I imagine it would not survive 150 C. See also JACS 2014, 136, 1226; Synlett 2015, 26, 2462; Organic Process Research & Development 2018, 22, 1672, and others. At least a few of these references should be included in the revision before publication.

Figure 1 entry 2 – what does “dis.” mean after Fe(CO)₅? Should be spelled out.

Figure 2A, the curve for 200 C actually goes above 100% conversion at about 30 min – should be revised.

Figure 3 caption starts with “Scope of neat alkenes...” but it would be more accurate to say “Scope of products from neat isomerization reactions...” because I see 10 and 11 and 13 with no alkene!

Top of page 8, “The Ru source was selected on the basis of its low price and bench stability (Fig. S5) since Ru₃(CO)₁₂ must be handled under nitrogen” seems like it better belongs in the discussion of Figure 2.

Lower part of page 8 – please show the structure of 25.

Figure 4B – what is KY? It needs to be defined clearly.

Figure 6 – the CV data need additional details. The upper traces are before reaction and the lower ones after? This needs to be specified. Also the direction of the CV traces needs to be specified. For example in the lower left one, why is there a third trace at the bottom? At what potential was the CV experiment started? The electrolyte present should also be identified here in the caption – I know it is in the experimental, but in the caption is needed also. In the experimental, it is stated that an equal volume of MeCN was added. MeCN is well-known to coordinate readily to Ru, so probably what we are seeing in the CV experiments are Ru(nitrile) species – please cite the relevant literature on electrochemistry of Ru(II)(CH₃CN)_n complexes and discuss. Furthermore, a CV feature is not assigned to one oxidation state, but to a redox couple like II/III, so the features need to be identified as such.

How are the authors explaining three reduction waves for RuCl₃? III/II, II/I and I/0?

My strong suggestion is to repeat these experiments with higher concentrations of Ru so that the CV features are more distinct.

Also in Figure 6, the arrow pushing and assignment of a Ru(II)(allyl)(hydride) are not correct. Please delete the arrows and reassign the formal oxidation state of the allyl complex as drawn to IV.

There is no evidence that I can see for actual chain-walking (which I understand to mean that the metal stays on the substrate the entire time) –please delete “chain walking” from Figure 6C, and also from the text right before conclusions.

Figure S11 – the signals for the tris and tetrakis alkene complexes need to be made visible! And the expected masses for the species need to be calculated with the same number of significant figures as the instrument gives. The Figure is really unacceptable as is.

How is KIE 1.45 obtained from Fig S15? The SI should detail what part of the curves are being used, and some explicit analysis of the data to justify 3 significant figures should be given.

The manuscript by Leyva–Pérez and co-coworkers, entitled “*Parts–per–million of ruthenium catalyze the selective chain–walking reaction of terminal alkenes*”, presents a well-conceived idea based on their and others prior work that low level amounts of Ru can catalyze chain walking under neat substrate conditions starting from a variety of Ru sources. The work is well-executed and presents a wealth of experimental data that are, in generally, carefully interpreted. As such, I think this will be a valuable paper in *Nature Communications*. That said, I have some (i) science suggestions, and (ii) some other writing and more minor suggestions, detailed below.

Suggestions for the Science

- (i) On p. 4, at the end of the Introduction just above “Results”, the authors should replace the broad, unspecific “...this approach...” with as specific of a hypothesis that they can, including what main observables it will tested by. Why? The more specific the hypothesis, the greater its “power”; the easier it is to test quantitatively and potentially disprove. A main alternative hypothesis there that the authors have tested would be good as well (see Platt, *Science* **1964**, *146*, 347 for more on these key points).
- (ii) The work makes a good effort at trying to identify the catalyst and if it is “homogeneous or heterogeneous”. That said, the paper should find and cite some of that classic, critical work on that “homogeneous or heterogeneous catalysis” problem as key background for readers such as the broad *Nature Communications* audience.
- (iii) Specifically, *catalyst poisoning experiments* the authors will find in that literature hold promise of being able to (a) confirm the precise level of catalyst present, and (b) support a monomeric mono-hydride Ru-H vs, for example, vs a RuH₂, vs even possibly a “(Ru-H)₄” catalyst or catalyst resting state—see more on these points below.
 - a. An issue here finding a poison that has a binding constant strong enough so that it will bind at the very low concentrations—maybe EDTA?
- (iv) Also, Ru-H can in principle be quantitated by its reaction with CCl₄, Ru-H + CCl₄ —> CHCl₃ (Orbitrap MS?) + Ru-Cl. Note that this experiment can in principle test for a neutral Ru^{II}(H)₂ catalyst vs the perhaps less likely cationic {H-Ru^{II}}⁺ proposed, unless the detection limits make this a very difficult experiment to carry out in the present, low Ru-loadings system.
- (v) Hugely interesting and telling in this work is that all of the completely different 10 Ru-starting complexes in Table S1 give, it appears, *pretty much the same catalyst and pretty close selectivity*. As such, I recommend putting Table S1 in the main text and giving those key results more emphasis and greater discussion.
- (vi) More specifically, the above “same apparent catalyst” finding means that at the 150 °C reaction temperature most if not all of the *strong binding* ligands come off of all these complexes in excess olefin ligand—as the authors briefly postulate in the “Ru(1 = olefin)₄” *briefly mention* on p. 12, 4 lines from the bottom.
- (vii) Along these same lines, the author can combine all their data for “*per-olefin*” (as they cite it, p. 13, line 5) and Ru(II)-H into a more specific working hypothesis for going forward for the active catalyst, basically “HRu^{II}(olefin)₂⁺” (see below), working from their evidence.

- a. Note the “+” charge is not given by the authors but should be and, then, the issue of getting a common counter cation from all the 10 different precursors comes up—something that argues pretty strongly against any “ $\text{HRu}^{\text{II}}(\text{olefin})_x^{+}$ ” catalyst, one can argue.
- (viii) Some additional points here. One is that a Ru(II) d^8 complex is likely 4-coordinate, square planar, so not the “ $\text{Ru}(\mathbf{1} = \text{olefin})_4$ ” the authors suggest in their paper without making the implied oxidation state clear. More likely in any event is that “ $\text{HRu}^{\text{II}}(\text{olefin})_3^{+}$ ” *would be a resting state*. Better one can argue is that neutral “ $\text{H}_2\text{Ru}^{\text{II}}(\text{olefin})_2$ ” is a more likely, d^8 , 4-coordinate, square planar, catalyst resting state.
- a. Actually, given Halpern’s work on the mechanism of Wilkinson’s catalyst (and other studies in the literature showing that d^8 , 4-coordinate, square planar complexes are often not very reactive) teach that the above 16 electron species are probably NOT the catalyst by literature precedent. Instead, 14 electron species such as “ $\{\text{HRu}^{\text{II}}(\text{olefin})_2\}^{+}$ ” or “ $\text{H}_2\text{Ru}^{\text{II}}(\text{olefin})_1$ ” have the much stronger precedent for actually being highly active catalysts.
 - b. Another important issue is that the authors imply what they see spectroscopically as Ru(CO) by IR are directly connected to the active catalyst, namely the 2143 cm^{-1} Ru-CO species seen and noted on p. 13, line 3 from the bottom. Halpern’s work with Wilkinson’s catalyst (*op. cit.*) disproves that general assumption. Additionally, the authors would need to have kinetics of the evolved active catalyst system (vs just its formation) to answer this challenging mechanistic question.
 - c. Making matters kinetically worse here is that Halpern also has key work showing that one cannot distinguish OFF-path from ON-path species that build up by any kinetics measurement under any steady-state conditions. Pre- and post-steady state kinetics are needed. The authors will want, therefore, to not claim the Ru-species they are detecting spectroscopically at 2143 cm^{-1} is the active catalyst—probably not. Those measurements do, however, offer valuable hypotheses for the active catalyst that the authors can and should try to disprove (Platt, *op. cit.*). Again, the literature cited is pretty clear on this point.
- (ix) The key HPLC-Orbitrap MS results in Figure S11 deserve to be presented in the main text and discussed more. First, there is not just the “ $\text{Ru}(\mathbf{1} = \text{olefin})_4$ ” mentioned and emphasized in the main text, but also $\text{Ru}(\mathbf{1} = \text{olefin})_2$, and $\text{Ru}(\mathbf{1} = \text{olefin})_3$ species cited there.
- a. Additionally, the MS peaks are weak (understandably, given the very low concentration of Ru). But, the specific chemical species and their molecular weights (m/e values), corresponding to the claimed $\text{Ru}(\mathbf{1} = \text{olefin})_2$, and $\text{Ru}(\mathbf{1} = \text{olefin})_3$ and the emphasized $\text{Ru}(\mathbf{1} = \text{olefin})_4$ species, need to be given with specific m/e values and also showing the calculated isotope envelopes in comparison to the observed MS data. Those should be definitive, at least if the S/N allows. The MS data at present are qualitative and unconvincing.
 - b. Also, can these species have Ru-H on them that would just increase the mass by 1? What about using D, and seeing a Ru-D of 2 additional mass units. That would be pretty convincing.

- (x) The Ru-Ligand thermochemistry in the conversation of all of the different Ru precatalysts, to an apparently common catalyst, is interesting, so the authors should at least think some about this. Why? Many (most?) readers might think it is crazy that all the strong, often chelating ligands in the 10 complexes in Scheme S1 can come off thermodynamically and under the excess olefin conditions. But, probably this is allowed / downhill (“spontaneous”) thermodynamically, as I detail further next.
- With for example 3 Ru(II)-olefin bonds of perhaps 25-35 kcal/mole and formation of a Ru-H (~70 kcal/mol) and its trans influence and trans effect should be enough to labilize the strong bonding ligands both thermodynamically and kinetically in the 10 Ru precursors (that, by these back-of-the-envelope calculations, cannot be have more Ru-Ligand total bond / binding energy for *any* of the precursors in Table S1 stronger than $(3 \times 35) + 70 = 175$ kcal/mole, in the postulate that the 10 precursors lose all their initial ligands, save maybe any Hydrides. Hence, I suggest the authors: (a) emphasize their very interesting data for a “common catalyst” more, and (b) also point out more clearly the now only lightly mentioned, key “...similar selectivity...” finding on p. 5, 6 lines from the bottom. (c) Talk about the thermochemistry some, too, *as I assure you that people that have worked with the many, pretty stable complexes in Scheme S1 will not like the idea that all the ligands in these precatalysts can (all) come off!*
 - Also, there is literature the authors should find and cite that, for example, where pincer and other, seemingly unremovable ligands can come off under certain catalysis conditions and at higher temperatures. I’ve seen papers on this point over the years, and good papers on this point this past year or two.
- (xi) The above and the literature suggests another hypothesis, hinted at above, that the authors should at least consider and also list among the working hypothesis going forward for the active catalyst. That hypothesis is that *neutral* $(\text{H})_2\text{Ru}^{\text{II}}(\text{olefin})_x$ and its possible tetramer, $\{\text{H}\}_2\text{Ru}^{\text{II}}(\text{olefin})_2\}_4$, that has precedent of sorts in an isoelectronic $\{\text{HIr}(\text{olefin})_2\}_4$ complex, is involved either as a catalyst resting state or an actual catalyst. See also the refers therein as I recall for somewhat analogous Ru_4 complexes suggested from a literature survey (Yih, K-H. et al. *Inorganic Chemistry*, **2012**, *51*, 3186-3193).
- Note the authors interesting finding that a *closed system* (that cannot lose H_2 generated in the synthesis from an $[(\text{COD})\text{Ir}^{\text{I}}\text{Cl}]_2$ precursor and R_3BH^-) *is required* for the successful synthesis of $\{\text{HIr}(\text{olefin})_2\}_4$. Not sure if this latter point and paper is of value to the present study, but it might be? (Laxson, W. W., et. al. *Inorg. Chim Acta.*, **2015**, *432*, 250-257).
 - An underlying issue here is that organometallic chemists know that M-H tend to do olefin isomerization, while MH_2 tend to do reductions of the double bond—that are not the reaction observed. This fact points *some* to a mono-hydride, Ru-H—but maybe not an in the present system, as there is no extra source of H other than from the olefin. More literature searching on this point, and if a $\text{Ru}(\text{H})_2(\text{olefin})_x$ species would make for a very labile, active, sterically small and thereby good catalyst, is needed by the authors.
- (xii) A couple controls the authors should do (if I did not miss them in the paper; apologies if I did) is do a few key experiments under Ar (and not O_2): do the conversion, selectivity, or any other key results that might be sensitive to O_2 change? What I am

thinking here is that O₂ could be reoxidizing any Ru(0)_n formed, thereby keeping the Ru in the apparently desired Ru(II) oxidation state.

- (xiii) Finally, the sigmoidal kinetics the authors see will likely be fit by a classic 2-step mechanism in the literature for catalyst evolution and growth processes in general: $A \rightarrow B$, then $A + B \rightarrow 2B$. If so, then see the very telling example of how Bergman and collaborators used a fit to this seemingly too simple kinetic scheme to deduce what B, the catalyst, actually is, as well as to confirm that A is the precatalyst (in your case, A should equal the Ru-catalyst precursor). The hugely interesting thing here is that if the above 2 step mechanism fits the data (I'm almost sure it will), then you will have direct kinetics evidence *that B is an autocatalyst*: that is, direct kinetics evidence that B makes more catalyst autocatalytically from something like $A = \text{Ru}(\text{precat}) + B = \text{H-Ru} \rightarrow 2 B = \text{Ru-H}$, as one educated guess here that fits your system and make chemical sense—one good mechanistic hypothesis, then. (See Smith, S. E.; Bergman, R. G.; et al. *J. Am. Chem. Soc.* **2008**, *130*, 1839-1841. Study especially the pseudo-elementary step treatment in this example in the SI that will be close to your own case, because all the other reagents are in large excess, and hence effectively constant, vs the precatalyst A and catalyst B concentrations—so just like in the Bergman system where the other reagents are in ≥ 10 -fold excess, as I recall. Study how these authors figured out the precise composition of both B as well as A—and then how that then gave them the detailed mechanism when they were otherwise totally stuck on the mechanism.)
- (xiv) I realize that much of my comments above are pretty kinetics and mechanistic, some likely beyond paper's main intentions or space limitations. If so, summarize best data and main highlights for the active catalyst, make the needed edits for the above comments, do the additional controls, and then go after the issues I have raised in a subsequent, more mechanistic paper.

More Minor Comments

- (1) A couple places need more experimental detail on the precise conditions used:
- Is “mol%” in the Table 4 Ru mol %, as I think? Please clarify this.
 - The conditions in Figure 6 (a) and (b), and their overlap or not with the catalytic conditions, are somewhat unclear and could therefore use some additional details there.
 - P. 17, under Reaction Procedures, please give us the precise amounts and total (milli-micro?) mols of MeOH and CHCl₂ that wind up in the final reaction solution. One point here, while slower than the aforementioned reaction with CCl₄, the reaction $M-H + CH_2Cl_2 \rightarrow CH_3Cl + M-Cl$ is known (but considerably slower as I recall). Nevertheless, a control changing say tripling the [CH₂Cl₂], and seeing if doing so has any effect, would be useful here.
- (2) Adding some **Bold Subheadings** to the Results would help the organization of the paper, and hence help the average reader, of this paper. The current presentation is a bit run-on at present in the Results—but still largely logically laid out and presented. Just a thought for the authors and Editor to consider.
- (3) The English is pretty good—far superior to anything I would be able to write in the author's native language of Spanish!, But, the paper has a few places that need the help of an English-language editor. A few to help with that task are:

- a. P. 2, Abstract “a plethora”—replace this by a specific number, namely the 9 broad reactions in Figure 5. Also on p. 2 add “additional” to read “additional ligands (as ligands are present); “replace “but just heating” by “only” heating; and add quantities to read “kilogram quantities”.
- b. P. 2, “internal alkenes have” (not had);
- c. P. 3, the sentence in line 3 of “selective for the product...and waste generating” is confusing; please rewrite this sentence.
- d. P. 5, “a ~~perfect~~-Arrhenius behavior”—delete perfect, as that error free case can never exist experimentally.
- e. P. 10, “is ~~low~~, although appreciable” makes little sense; say perhaps “is less, although appreciable”;
- f. *Very nice experiments in Figure 5, by the way, using the as prepared isomerized olefin without any needed purification!*
- g. Figure 6, part C caption: **Proposed** Reaction Mechanism.

Finally, my compliments to the authors for an interesting, detailed, extensive experimental study and thoughtful, careful reporting of their work. I look forward to reading the final paper in Nature Communications, assuming the other referees and Editor agree with my assessment.

MINISTERIO
DE ECONOMÍA
Y COMPETITIVIDAD

CSIC
CONSEJO SUPERIOR DE INVESTIGACIONES CIENTÍFICAS

UNIVERSITAT
POLITÈCNICA
DE VALÈNCIA

INSTITUTO DE
TECNOLOGÍA
QUÍMICA

Dr. Antonio Leyva Pérez
Tenured scientist
Instituto de Tecnología Química UPV-CSIC
Avda. de los Naranjos s/n,
Universidad Politécnica de Valencia
46022 VALENCIA, SPAIN

First of all, let us think to the Reviewers for their very professional reviews and the nice comments. A point-by-point answer to all their concerns, follows.

Reviewer #1:

The authors make a convincing case for a more cost-effective transition metal alkene isomerization catalyst, and they provide a nice example. The Ru(methallyl)(COD) complex is devoid of higher molecular weight phosphines or strongly-held CO ligands that may in fact inhibit the catalysis. The results shown are highly significant and definitely deserving of publication after some revision, but the revisions are absolutely needed before publication. TON of 1×10^8 per h is quite amazing! But of course this is at 200 C. The use of supported materials and thorough characterization is notable. Also significant is that the very low amounts of ruthenium allow for subsequent catalyzed transformations.

We truly thank the Reviewer for the nice comments.

There are several downsides however. Higher temperatures such as 150 or 200 C do lead to a loss of E/Z selectivity because there is no kinetic control, but if that loss is tolerable in the application then it does not matter. A number of the Figures need some work, some key literature is not cited, but as far as I can see, with the exception of some electrochemical work, only editorial changes would be needed to make the paper suitable for publication, but as the paper is now it is not acceptable.

Thanks for the comments. A temperature >150 °C is required to generate, in-situ, the true Ru catalyst. The mixture of isomers at then end, mainly trans, it is not a problem for many applications, and it does not compare negatively with other more complex synthetic procedures of internal alkenes where mixtures of isomers are obtained anyway (olefin metathesis, Wittig-type reactions....).

Following the Reviewer's recommendations, the whole scholarship presentation of the manuscript has been revised, including Figures, and 15 new references are now included in the manuscript.

It is very interesting that the system does not affect internal double bonds (Fig S6B) because it is so active with terminal ones. Can the authors comment? If a tris or tetrakis (terminal alkene) complex is the active catalyst, it could be quite bulky thanks to the presence of several alkenes.

What about some (admittedly somewhat speculative) calculations in support of pi allyl hydride mechanism on Ru(II)(alkene)_n species?

Thanks for the insightful comments. Following the Reviewer's suggestions, we have performed density functional theory (DFT) calculations to further understand the mechanism of the Ru-catalyzed isomerization. The results are included in the new Figure S18. It can be seen there that the isomerization of the Ru atom in a model terminal alkene (1-butene) is energetically favoured in, at least, 13 kcal·mol⁻¹, and that Ru(II) stabilizes twice the p-allyl intermediate than Ru(I). These results have been included in the main text with a new sentence.

What is the purity of the feedstocks? There are no details mentioned, unless I am missing something. I ask because over time, alkenes with allylic positions suffer some oxidation. If the feedstocks are not purified in any way, this would be a plus for the manuscript.

Thanks again for the comment. The starting alkenes have not been purified in any case, and they come from very different commercial houses or prepared by us. Following the Reviewer's suggestion, we have included a new sentence in the main text of the manuscript and also in the experimental section commenting on this issue.

There are no references to significant work on alkene chain walking catalysis from the group of Sigman – e.g. JACS 2020, 142, 10516; J Org Chem 2014, 79, 11841; and beautiful work in Nature 2014, 508, 340.

Sorry for not including these relevant references before, they are now included as new reference numbers 28-30.

The authors have not cited any of the considerable work by Grotjahn on alkene isomerization, including reactions that finish in less than 1 h at ambient temperature using only 500 ppm of catalyst, and taking 24 h with 100 ppm – I had to refresh my memory on this, but see in particular JACS 2012 134, 10357. Clearly the Grotjahn catalyst is more expensive, but the fact that it operates at ambient temperatures is useful – it even works below zero deg C! – see ACS Catal. 2020, 10, 15250. I imagine it would not survive 150 C. See also JACS 2014, 136, 1226; Synlett 2015, 26, 2462; Organic Process Research & Development 2018, 22, 1672, and others. At least a few of these references should be included in the revision before publication.

Again, our apologies for not including these amazing works in the manuscript. Three of these references are now included in the main text as the new reference numbers 7-9, and we could not include all of them because they exceeded the number limit (counting with those suggested by Reviewer 2).

Figure 1 entry 2 – what does “dis.” mean after Fe(CO)₅? Should be spelled out.

Figure 2A, the curve for 200 C actually goes above 100% conversion at about 30 min – should be revised.

Thanks again for the comments. We have deleted “dis” after Fe(CO)₅, since it means “dissolved” and the experimental procedure is already commented in the proper section.

Figure 3 caption starts with “Scope of neat alkenes...” but it would be more accurate to say “Scope of products from neat isomerization reactions...” because I see 10 and 11 and 13 with no alkene! Top of page 8, “The Ru source was selected on the basis of its low price and bench stability (Fig. S5) since Ru₃(CO)₁₂ must be handled under nitrogen” seems like it better belongs in the discussion of Figure 2.

Thanks again. The Figure 3 caption and the sentence have been changed accordingly (the old Fig. S5 is now Fig. S2, and the rest of Figures in the SM has been re-numbered).

Lower part of page 8 – please show the structure of 25.

Compound 25 is a mixture of the different isomers, and it is commercialized in that form. We think that this delocalized structure is the best way to show it.

Figure 4B – what is KY? It needs to be defined clearly.

Sorry for this mistake, the Reviewer is right, KY meaning should have been clearly defined in the text. KY refers to zeolite Y exchanged with K by Na. The clarification is now included in the main text and also in the Figure 4 caption.

Figure 6 – the CV data need additional details. The upper traces are before reaction and the lower ones after? This needs to be specified. Also the direction of the CV traces needs to be specified. For example in the lower left one, why is there a third trace at the bottom? At what potential was the CV experiment started? The electrolyte present should also be identified here in the caption – I know it is in the experimental, but in the caption is needed also. In the experimental, it is stated that an equal volume of MeCN was added. MeCN is well-known to coordinate readily to Ru, so probably what we are seeing in the CV experiments are Ru(nitrile) species – please cite the relevant literature on electrochemistry of Ru(II)(CH₃CN)_n complexes and discuss. Furthermore, a CV feature is not assigned to one oxidation state, but to a redox couple like II/III, so the features need to be identified as such.

Thanks again for the insightful comments. The Reviewer is right, this Figure needs more work on it. Following the Reviewer’s comments, we have included now, in both the caption and the main text, the details about the CV traces, the potential at which the experiment started and the electrolyte present. Besides, we have marked with an asterisk the peaks corresponding to nitrile species, and we have cited two new references (numbered as 43 and 44) covering relevant literature on electrochemistry of Ru(II)(CH₃CN)_n complexes, and a discussion on them is now included in the main text. The CV features are now identified as couples, sorry for this mistake.

How are the authors explaining three reduction waves for RuCl₃? III/II, II/I and I/0?

MINISTERIO
DE ECONOMÍA
Y COMPETITIVIDAD

CSIC
CONSEJO SUPERIOR DE INVESTIGACIONES CIENTÍFICAS

UNIVERSITAT
POLITÈCNICA
DE VALÈNCIA

INSTITUTO DE
TECNOLOGÍA
QUÍMICA

We have not a clear explanation for this, but we have to conclude that in a first approach.

My strong suggestion is to repeat these experiments with higher concentrations of Ru so that the CV features are more distinct.

Also in Figure 6, the arrow pushing and assignment of a Ru(II)(allyl)(hydride) are not correct. Please delete the arrows and reassign the formal oxidation state of the allyl complex as drawn to IV.

Thanks again for the comments, we have deleted all arrows and reassigned them properly.

There is no evidence that I can see for actual chain-walking (which I understand to mean that the metal stays on the substrate the entire time) –please delete “chain walking” from Figure 6C, and also from the text right before conclusions.

We have now removed “chain-walking” from Figure 6C and the text, and substituted it by “isomerization”. This has also been done during the introduction and the whole text, where additional “chain-walking” terms appeared.

Figure S11 – the signals for the tris and tetrakis alkene complexes need to be made visible! And the expected masses for the species need to be calculated with the same number of significant figures as the instrument gives. The Figure is really unacceptable as is.

The Reviewer is absolutely right, sorry for these mistakes. We have now included the expected masses with four decimals. Regarding the intensity of the signals, please notice that we are in the detection limit of the instrumentation, thus amplification is not possible without too much noise. However, we still think that the Figure is informative and we have kept it.

How is KIE 1.45 obtained from Fig S15? The SI should detail what part of the curves are being used, and some explicit analysis of the data to justify 3 significant figures should be given.

Many thanks for the comments. The KIE is obtained with the initial rates of the reactions, obtained by linear regression of the initial points in the linear part of the curve. For the deuterated experiment (blue line), it can be observed an induction time, which is not considered for the calculation of the initial rate. The linear regression for both reactions gave us the second decimal with reasonable precision, however, for the sake of accurateness, we have now included in the text this second decimal as a potential error: 1.4(5). Following the Reviewer’s suggestion, we have included a new sentence in the experimental part of the manuscript to clarify this point.

Reviewer #2:

The manuscript by Leyva–Perez and co-coworkers, entitled “*Parts–per–million of ruthenium catalyze the selective chain–walking reaction of terminal alkenes*”, presents a wellconceived idea based on their and others prior work that low level amounts of Ru can catalyse chain walking under neat substrate conditions starting from a variety of Ru sources. The work is well-executed and presents a wealth of experimental data that are, in generally, carefully interpreted. As such, I think this will be a valuable paper in *Nature Communications*. That said, I have some (i) science suggestions, and (ii) some other writing and more minor suggestions, detailed below.

We really appreciate the nice comments of the Reviewer.

Suggestions for the Science

(i) On p. 4, at the end of the Introduction just above “Results”, the authors should replace the broad, unspecific “...this approach...” with as specific of a hypothesis that they can, including what main observables it will tested by. Why? The more specific the hypothesis, the greater its “power”; the easier it is to test quantitatively and potentially disprove. A main alternative hypothesis there that the authors have tested would be good as well (see Platt, *Science* **1964**, *146*, 347 for more on these key points).

Many thanks for this comment. Following the Reviewer’s suggestion, we have changed “this approach” for the more accurate hypothesis: “Thus, it was envisioned here that tiny amounts of Ru species formed from a variety of Ru sources may catalyse, very efficiently, the alkene isomerization reaction.” Since this sentence already includes the obvious variable to test (different Ru sources and tiny amounts), we have not included more details at this introductory point.

(ii) The work makes a good effort at trying to identify the catalyst and if it is “homogeneous or heterogeneous”. That said, the paper should find and cite some of that classic, critical work on that “homogeneous or heterogeneous catalysis” problem as key background for readers such as the broad *Nature Communications* audience.

The Reviewer is right, and we have introduced a new sentence and two new references (numbers 33 and 34) in the main text, during the section of recoverable catalysts, to comment on this general issue in catalysis.

(iii) Specifically, *catalyst poisoning experiments* the authors will find in that literature hold promise of being able to (a) confirm the precise level of catalyst present, and (b) support a monomeric mono-hydride Ru-H vs, for example, vs a RuH₂, vs even possibly a “(Ru-H)₄” catalyst or catalyst resting state—see more on these points below.

a. An issue here finding a poison that has a binding constant strong enough so that it will bind at the very low concentrations—maybe EDTA? (iv) Also, Ru-H can in principle be quantitated by its reaction with CCl₄, Ru-H + CCl₄ → CHCl₃ (Orbitrap MS?) + Ru-Cl. Note that this experiment can in principle test for a neutral RuII(H)₂ catalyst vs the perhaps less likely cationic {H-RuII}⁺ proposed,

unless the detection limits make this a very difficult experiment to carry out in the present, low Ru-loadings system.

Many thanks for these insightful comments. Following the Reviewer's suggestions, we have carried out new experiments with EDTA, CCl₄ and, as an additional radical quencher, TEMPO, to get more information about the nature of the true Ru catalytic species. The results are shown in the new Figure S17. It can be seen there that only EDTA produces a significant decrease of the reaction rate, while CCl₄ and TEMPO do not. These results point to a cationic Ru species as plausible intermediate. We have included a new sentence in the main text to comment on these new results.

(v) Hugely interesting and telling in this work is that all of the completely different 10 Ru-starting complexes in Table S1 give, it appears, *pretty much the same catalyst and pretty close selectivity*. As such, I recommend putting Table S1 in the main text and giving those key results more emphasis and greater discussion.

The Table with the different starting Ru complexes which finally give the same catalytic results is included in Figure 2 of the manuscript and commented in the main text.

(vi) More specifically, the above “same apparent catalyst” finding means that at the 150 °C reaction temperature most if not all of the *strong binding* ligands come off of all these complexes in excess olefin ligand—as the authors briefly postulate in the “Ru(1= olefin)₄” *briefly mention* on p. 12, 4 lines from the bottom.

Thanks again for the comments. Yes, the Reviewer is right, and the starting ligands (whatever they are) break under the reaction conditions. A new experiment has been carried out to confirm this finding (see below in a related question of the Reviewer).

(vii) Along these same lines, the author can combine all their data for “*per-olefin*” (as they cite it, p. 13, line 5) and Ru(II)-H into a more specific working hypothesis for going forward for the active catalyst, basically “HRu(II)(olefin)₂⁺” (see below), working from their evidence.

a. Note the “+” charge is not given by the authors but should be and, then, the issue of getting a common counter cation from all the 10 different precursors comes up—something that argues pretty strongly against any “HRu(II)(olefin)_x⁺” catalyst, one can argue.

We have not included the “+” symbol in the postulated Ru-per-olefin intermediate complex since its structure is not clear at this point and, moreover, we do not know the counteranion of this intermediate. However, as the Reviewer has pointed out, the cationic nature of this Ru intermediate is counterintuitively observed in the “Ru-per-olefin” nomenclature, and we prefer leaving this inaccurate but not erroneous naming.

(viii) Some additional points here. One is that a Ru(II) d₈ complex is likely 4-coordinate, square planar, so not the “Ru(1 = olefin)₄” the authors suggest in their paper without making the implied oxidation state clear. More likely in any event is that “HRu(II)(olefin)₃⁺” *would be a resting state*.

Better one can argue is that neutral “H₂Ru(olefin)₂” is a more likely, d₈, 4-coordinate, square planar, catalyst resting state.

a. Actually, given Halpern’s work on the mechanism of Wilkinson’s catalyst (and other studies in the literature showing that d₈, 4-coordinate, square planar complexes are often not very reactive) teach that the above 16 electron species are probably NOT the catalyst by literature precedent. Instead, 14 electron species such as “{HRu(olefin)₂}+” or “H₂Ru(olefin)₁” have the much stronger precedent for actually being highly active catalysts.

b. Another important issue is that the authors imply what they see spectroscopically as Ru(CO) by IR are directly connected to the active catalyst, namely the 2143 cm⁻¹ Ru-CO species seen and noted on p. 13, line 3 from the bottom. Halpern’s work with Wilkinson’s catalyst (*op. cit.*) disproves that general assumption. Additionally, the authors would need to have kinetics of the evolved active catalyst system (vs just its formation) to answer this challenging mechanistic question.

c. Making matters kinetically worse here is that Halpern also has key work showing that one cannot distinguish OFF-path from ON-path species that build up by any kinetics measurement under any steady-state conditions. Pre-and post-steady state kinetics are needed. The authors will want, therefore, to not claim the Ruspecies they are detecting spectroscopically at 2143 cm⁻¹ is the active catalyst—probably not. Those measurements do, however, offer valuable hypotheses for the active catalyst that the authors can and should try to disprove (Platt, *op. cit.*). Again, the literature cited is pretty clear on this point.

We really thank the Reviewer for her/his insightful and generous comments on the more difficult part of the study, i.e. the reactive mechanism and the Ru active species. Following the Reviewer’s rationalization, we have included new sentences in the manuscript that summarizes most of the Reviewer’s suggestions, particularly the possible inactivity of Ru d₁₆ species and the Ru-CO intermediate detected by FT-IR. For the latter, the Ru(II)-CO intermediate detected has been assigned as a labile species during reaction, but not as the true active species. A reference from Halpern-s work (new Ref. 51) has been added. When commenting the reaction mechanism, the following sentence has been added: “It must be notice here that, at this point, it is difficult to exactly know the structure of the active complex, however, according to previous results, a fully coordinated, presumably square planar d₁₆ per-alkene RuII complex is plausible a resting state while a more reactive d₁₄ tris-alkene RuII would be more favourable for catalysis⁵¹.”

(ix) The key HPLC-Orbitrap MS results in Figure S11 deserve to be presented in the main text and discussed more. First, there is not just the “Ru(1 = olefin)₄” mentioned and emphasized in the main text, but also Ru(1 = olefin)₂, and Ru(1 = olefin)₃ species cited there.

a. Additionally, the MS peaks are weak (understandably, given the very low concentration of Ru). But, the specific chemical species and their molecular weights (m/e values), corresponding to the claimed Ru(1 = olefin)₂, and Ru(1 = olefin)₃ and the emphasized Ru(1 = olefin)₄ species, need to be given with specific m/e values and also showing the calculated isotope envelopes in

comparison to the observed MS data. Those should be definitive, at least if the S/N allows. The MS data at present are qualitative and unconvincing.

b. Also, can these species have Ru-H on them that would just increase the mass by 1? What about using D, and seeing a Ru-D of 2 additional mass units. That would be pretty convincing.

Thanks again for the useful comments. The Reviewer is right, we have included now the expected m/z values for the Ru complexes with four decimals, as the experimental technique gives the results. In this way, it does not appear that the complexes are protonated, rather they tend to lose hydrogen atoms during the measurement. This makes the great idea of using deuterium suggested by the Reviewer, unsuitable here. As the Reviewer points out, the signal to noise difference is very small, too small to get conclusions about the isotopic envelope. With all these drawbacks in mind, we prefer to leave the Figure in the SM rather than translating it into the manuscript.

(x) The Ru-Ligand thermochemistry in the conversation of all of the different Ru precatalysts, to an apparently common catalyst, is interesting, so the authors should at least think some about this. Why? Many (most?) readers might think it is crazy that all the strong, often chelating ligands in the 10 complexes in Scheme S1 can come off thermodynamically and under the excess olefin conditions. But, probably this is allowed / downhill (“spontaneous”) thermodynamically, as I detail further next.

a. With for example 3 Ru(II)-olefin bonds of perhaps 25-35 kcal/mole and formation of a Ru-H (~70 kcal/mol) and its trans influence and trans effect should be enough to labilize the strong bonding ligands both thermodynamically and kinetically in the 10 Ru precursors (that, by these back-of-the-envelope calculations, cannot be have more Ru-Ligand total bond / binding energy for *any* of the precursors in Table S1 stronger than $(3 \times 35) + 70 = 175$ kcal/mole, in the postulate that the 10 precursors lose all their initial ligands, save maybe any Hydrides. Hence, I suggest the authors: (a) emphasize their very interesting data for a “common catalyst” more, and (b) also point out more clearly the now only lightly mentioned, key “...similar selectivity...” finding on p. 5, 6 lines from the bottom.

(c) Talk about the thermochemistry some, too, *as I assure you that people that have worked with the many, pretty stable complexes in Scheme S1 will not like the idea that all the ligands in these precatalysts can (all) come off!*

b. Also, there is literature the authors should find and cite that, for example, where pincer and other, seemingly unremovable ligands can come off under certain catalysis conditions and at higher temperatures. I’ve seen papers on this point over the years, and good papers on this point this past year or two.

The Reviewer is right in that the use of practically any Ru complex, independently of the ligands present, leads to the same catalytic results (activity and selectivity) and could be surprising (if not offensive) for some Researchers in the field. Following the Reviewer’s suggestion, we have

included a new sentence in the main text emphasizing the same selectivity when starting from any Ru source (selectivity is a fingerprint of catalyst structure) and the rationalization with thermodynamic data provide by the Reviewer, as follows: “The fact that not only the catalytic activity but also the selectivity for a plethora of different Ru complexes is extraordinarily similar (see Figure 2 above), and that if one considers that the formation of three new Ru^{II}-olefin bonds of around 25-35 kcal·mol⁻¹ plus one Ru-H bond of ~70 kcal·mol⁻¹, with its *trans* effect, might be enough to labilize even the strong ligand bonds, strongly supports our original hypothesis that a common Ru species is formed under solvent-free heating conditions.”

Besides, we have carried out a new experiment to confirm the breaking of these ligands' bonds under reaction conditions. For that, we used the stable complex Ru(PPh₃)₃Cl₂ and we followed the reaction by in-situ NMR, to find that the complex is totally destroyed after a few minutes, with free phosphine as the main species of the mixture. The results are include in the new Figure S20 and a sentence has been added in the main text, as follows: “Indeed, in-situ ³¹P NMR experiments, in solution, showed how the PPh₃ ligands of the stable complex Ru(PPh₃)₃Cl₂ complex come off under reaction conditions (Fig. S20).”

Regarding previous references on the destabilization of stable metal complexes under reaction conditions to generate ligand-free catalysts, we have added the new reference 54.

(xi) The above and the literature suggests another hypothesis, hinted at above, that the authors should at least consider and also list among the working hypothesis going forward for the active catalyst. That hypothesis is that *neutral* (H)₂Ru^{II}(olefin)_x and its possible tetramer, {H)₂Ru^{II}(olefin)₂}₄, that has precedent of sorts in an isoelectronic {HIr(olefin)₂}₄ complex, is involved either as a catalyst resting state or an actual catalyst. See also the refers therein as I recall for somewhat analogous Ru₄ complexes suggested from a literature survey (Yih, K-H. et al. *Inorganic Chemistry*, **2012**, *51*, 3186-3193).

a. Note the authors interesting finding that a *closed system* (that cannot lose H₂ generated in the synthesis from an [(COD)IrCl]₂ precursor and R₃BH-) *is required* for the successful synthesis of {HIr(olefin)₂}₄. Not sure if this latter point and paper is of value to the present study, but it might be? (Laxson, W. W., et. al. *Inorg. Chim Acta.*, **2015**, *432*, 250-257).

b. An underlying issue here is that organometallic chemists know that M-H tend to do olefin isomerization, while MH₂ tend to do reductions of the double bond— that are not the reaction observed. This fact points *some* to a mono-hydride, Ru-H—but maybe not an in the present system, as there is no extra source of H other than from the olefin. More literature searching on this point, and if a Ru(H)₂(olefin)_x species would make for a very labile, active, sterically small and thereby good catalyst, is needed by the authors.

We deeply thank to the Reviewer for the very constructive suggestions on the reaction mechanism and true catalytic species. Following her/his advice, we have included a new sentence in the main text commenting on the possibility that the H₂-Ru complexes could be involved during reaction, as follows: “A Ru^{II}(olefin)_xH₂ and its possible tetramer,

{Ru^{II}(olefin)_xH₂}₂, must not be discarded as a potential catalyst resting state or actual catalyst on the basis of literature precedents with related {HR(olefin)₂}₄ complexes⁵²⁻⁵³. “ Besides, the 2 new references suggested by the Reviewer are now included in the manuscript, as new references number 52 and 53.

(xii) A couple controls the authors should do (if I did not miss them in the paper; apologies if I did) is do a few key experiments under Ar (and not O₂): do the conversion, selectivity, or any other key results that might be sensitive to O₂ change? What I am thinking here is that O₂ could be reoxidizing any Ru(0)_n formed, thereby keeping the Ru in the apparently desired Ru(II) oxidation state.

Thanks again for the insightful comments. We missed to comment on this factor on the manuscript. From our very beginning experiments, we early realized that the reaction atmosphere was irrelevant, since the results under nitrogen were exactly the same than under air. We chose the latter by practical reasons. Besides, oxidized Ru species are less active in the reaction (i.e. RuO₂, see Figure 2), what further supports that the re-oxidation of Ru seems to not play any role during the catalysis. Following the Reviewer’s suggestion, we have added a new sentence in the main text to comment on this: “The reaction proceeded with the same result when an atmosphere of N₂ was placed instead of air.”

(xiii) Finally, the sigmoidal kinetics the authors see will likely be fit by a classic 2-step mechanism in the literature for catalyst evolution and growth processes in general: A → B, then A + B → 2B. If so, then see the very telling example of how Bergman and collaborators used a fit to this seemingly too simple kinetic scheme to deduce what B, the catalyst, actually is, as well as to confirm that A is the precatalyst (in your case, A should equal the Ru-catalyst precursor). The hugely interesting thing here is that if the above 2 step mechanism fits the data (I’m almost sure it will), then you will have direct kinetics evidence *that B is an autocatalyst*: that is, direct kinetics evidence that B makes more catalyst autocatalytically from something like A = Ru(precat) + B = H-Ru → 2 B = Ru-H, as one educated guess here that fits your system and make chemical sense— one good mechanistic hypothesis, then. (See Smith, S. E.; Bergman, R. G.; et al. *J. Am. Chem. Soc.* **2008**, *130*, 1839-1841. Study especially the pseudoelementary step treatment in this example in the SI that will be close to your own case, because all the other reagents are in large excess, and hence effectively constant, vs the precatalyst A and catalyst B concentrations—so just like in the Bergman system where the other reagents are in ≥10-fold excess, as I recall. Study how these authors figured out the precise composition of both B as well as A—and then how that then gave them the detailed mechanism when they were otherwise totally stuck on the mechanism.)

Many thanks again for the extraordinarily useful suggestions. The Finke–Watzky model of course fits very well here and deserves a more profuse study, since not only will give us insights into the true active species but also will help to further improve the catalytic efficiency of the process. However, as the Reviewer states in the below comment, it is far beyond of the scope of this work and should be studied apart. We will do so. We have included the new reference

suggested by the Reviewer in the mechanistic discussion of the manuscript (new reference number 55).

(xiv) I realize that much of my comments above are pretty kinetics and mechanistic, some likely beyond paper's main intentions or space limitations. If so, summarize best data and main highlights for the active catalyst, make the needed edits for the above comments, do the additional controls, and then go after the issues I have raised in a subsequent, more mechanistic paper.

Many thanks for the suggestions. We have tried to include most of the Reviewer's suggestions along the new version of the paper, and following her/his advice, we will go soon for an in depth study on the mechanistic part, in a follow-up work.

More Minor Comments

(1) A couple places need more experimental detail on the precise conditions used:

a. Is "mol%" in the Table 4 Ru mol %, as I think? Please clarify this.

The Reviewer is right and the amount of Ru in reaction can be confused with the amount of Ru on each solid. We have now included a new sentence in the caption to clarify this, as follows: "The Ru wt% on each solid is indicated between brackets, mol% refers to the catalytic amount of Ru in reaction."

b. The conditions in Figure 6 (a) and (b), and their overlap or not with the catalytic conditions, are somewhat unclear and could therefore use some additional details there.

Yes, the experimental conditions in Figures 6A and 6B can be considered as in-situ experiments, since they give products, and a new sentence has been added to the caption of Figure 6 to clarify this point: "Notice that the experimental conditions in A) and B) are reasonable comparable with the standard catalytic reaction conditions here reported."

c. P. 17, under Reaction Procedures, please give us the precise amounts and total (milli-micro?) mols of MeOH and CH₂Cl₂ that wind up in the final reaction solution. One point here, while slower than the aforementioned reaction with CCl₄, the reaction M-H + CH₂Cl₂ → to CH₃Cl + M-Cl is known (but considerably slower as I recall). Nevertheless, a control changing say tripling the [CH₂Cl₂], and seeing if doing so has any effect, would be useful here.

The amount of either CH₂Cl₂ or MeOH solvent added is around 5-10 μl. Following the Reviewer's advice, this data is now included in the experimental. This amount corresponds to much more than 3 equivalents respect to Ru, and no rate decreasing was observed. Experiments with higher amounts of CH₂Cl₂ did not show any change in the final reaction outcome. It is true that CH₂Cl₂ rapidly goes off under the reaction temperature, however, we have to conclude here that no effect on the catalyst is occurring.

(2) Adding some **Bold Subheadings** to the Results would help the organization of the paper, and hence help the average reader, of this paper. The current presentation is a bit run-on at present in the Results—but still largely logically laid out and presented. Just a thought for the authors and Editor to consider.

Thanks for the comments. Following the Reviewer suggestion, we have added 5 new bold subheaded sections, in brief: Catalytic Ru sources, scope, solid catalysts, one-pot reactions and reaction mechanism. Of course, we leave the final decision to the Editor, however, let us saying that we agree with the Reviewer that the manuscript seems now easier to read.

(3) The English is pretty good—far superior to anything I would be able to write in the author's native language of Spanish!, But, the paper has a few places that need the help of an English-language editor. A few to help with that task are:

- a. P. 2, Abstract “a plethora”—replace this by a specific number, namely the 9 broad reactions in Figure 5. Also on p. 2 add “additional” to read “additional ligands (as ligands are present); “replace “but just heating” by “only” heating; and add quantities to read “kilogram quantities”.
- b. P. 2, “internal alkenes have” (not had);
- c. P. 3, the sentence in line 3 of “selective for the product...and waste generating” is confusing; please rewrite this sentence.
- d. P. 5, “a perfect Arrhenius behavior”—delete perfect, as that error free case can never exist experimentally.
- e. P. 10, “is low, although appreciable” makes little sense; say perhaps “is less, although appreciable”;
- f. *Very nice experiments in Figure 5, by the way, using the as prepared isomerized olefin without any needed purification!*
- g. Figure 6, part C caption: **Proposed** Reaction Mechanism.

Thanks for all these details that of course improve the final reading of the manuscript. All the changes proposed by the Reviewer are now included in the manuscript, and a deep revision of the English has been carried out.

Finally, my compliments to the authors for an interesting, detailed, extensive experimental study and thoughtful, careful reporting of their work. I look forward to reading the final paper in Nature Communications, assuming the other referees and Editor agree with my assessment

Many thanks again for the nice comments.

REVIEWER COMMENTS

Reviewer #1 (Remarks to the Author):

The authors have addressed most of my concerns, but not all, and some new concerns are raised below. I appreciate the work done, but here I will focus on the parts that need attention before publication.

Manuscript Figure 6. If the Ru-alkene pi complex is Ru(II), then the pi allyl hydride complex is Ru(IV). I asked that this be corrected but it was not. Moreover, the arrows that are trying to show atom movement should be deleted entirely; I apologize for not being clearer when I wrote that they should be corrected. Such arrows are supposed to show electron movement, and the present arrows do not, nor is it really feasible to show it.

Figure S11. The expected masses are given with enough significant figures, and most of the experimental masses are visible, but not for the peak 324. Most importantly, the “envelope” of all the peaks due to Ru isotopes (mostly) are still not visible. Please make enlargements so one can see all the peaks.

Figure S15 and added text in manuscript, experimental section, pp 23-24. The explanation alone is really not sufficient. Are the authors meaning the 30 to 60 min portion of the plot? If that is the case, it should be made clear.

Manuscript page 17. Rather than writing d¹⁶ or d¹⁴, one usually writes 16-electron or 14-electron – please change. The d superscript notation is used to describe the electron count of the metal, not the whole complex.

Figure S18. I thank the authors for including some DFT calculations, but definitely more needs to be done before publication. Please see points 1 to 3.

1. why Ru(I)? There is no explanation why this oxidation state was selected.
2. the calculations seem to show a “ligandless” Ru atom interacting with 1-butene, but what I want the authors to calculate is what they are actually proposing on the basis of experimental evidence such as mass spectrometry, for example with four or three bound alkenes, giving 16-electron or 14-electron species that are discussed in the text.

3. Customary presentation of computational results includes the following:

3a. Relevant details of the computational method, including basis sets

3b. Tables of atomic coordinates and E and G information for each.

3c. It would be nice to include some key bond length information.

Minor typos: Fig. 2 caption line 5, "the Ru valence state" should be "the Ru oxidation state" . Similar comment on next page, 4 lines from bottom. "low-valent" not "low-valence" . Last line of the paragraph, "unprecedented".

Per the editor's request, I have taken a look at the revised manuscript "*Parts-per-million of ruthenium catalyze the selective chain-walking reaction of terminal alkenes*", at the authors' thorough reply to my and referee #1's first round of comments, and at the Supporting Information. I find that the authors have carefully addressed my, and it would also appear much of referee #1's, comments and suggestions. I conclude that this paper is close to publication, following just a small number of additional suggestions, provided below.

Additional Suggestions

- (1) P. 13 in the Bold Heading and Figure 6 c: insert "proposed" to say "Proposed Reaction Mechanism". No mechanism is exactly correct to start, so that phrasing it this way will avoid criticisms later on when the mechanism gets updated by additional studies.
- (2) Most of p. 14, 15, and the ½ of p. 16 up to the "The mechanism of the isomerization reaction was studied...." is probably best placed in the SI, as it only indirectly bears on the mechanism. It would be better to just start the Reaction Mechanism discussion on p. 13 at the above sentence ("The mechanism of the isomerization reaction was studied...."), and then put the CV and the IR studies in Figure 6 into a separate, later, brief figure with much of the details in the SI. The revised Figure 6 can be just the "Proposed Reaction Mechanism". These changes will tighten the manuscript significantly from p. 13-16, and by doing so not have the mechanistic discussion detract from all the nice work presented p. 1-13 which is the main part of the paper.
- (3) P. 17, bottom: the authors should find some references and cite them (and check) the Ru-H and Ru-olefin BDEs that I quoted to them from memory and as ballpark numbers. These values are not particularly easy to find as they are spread out in the literature. But, look for them, starting with J. L. Beauchamp's *Chemical Reviews*, 1990, 90, 629-688, and papers, again just to start.
 - a. In Table 13 therein he lists in Ru-H = 234±21 kJ/mol (so 60 ± 5 kcal/mol) and Ru(II)-H⁺ as 172±13 kJ/mol (so 41± 3 kcal/mol)—note these are other-ligand-free binuclear species in the gas phase.
 - b. As for Ru(II)-Olefin BDEs, one place to start is Nechaev, Mikhail S. et al. *J Phys. Chem* **2004**, *108*, 3134-3142.
 - c. Again, be sure to search the rest of the literature for these Ru-ligand bond energies.
- (4) P. 4-5, I still think that Figure S3 in the Supporting Information should be moved to the main text, around p. 4-5, as seeing those structures there, then learning that all the complexes behave similarly under catalytic conditions at 150-200 °C, is a very powerful, telling part of the paper.

In summary, with attention to the above points and any additional comments from referee #1, I am happy to see the final, revised paper published in *Nature Communications*.

MINISTERIO
DE ECONOMÍA
Y COMPETITIVIDAD

UNIVERSITAT
POLITÈCNICA
DE VALÈNCIA

INSTITUTO DE
TECNOLOGÍA
QUÍMICA

Dr. Antonio Leyva Pérez
Tenured scientist / Distinguished Researcher
Instituto de Tecnología Química UPV-CSIC
Avda. de los Naranjos s/n,
Universidad Politécnica de Valencia
46022 VALENCIA, SPAIN

First of all, let us thanking again to the Reviewers for their very insightful reviews. A point-by-point answer to all their new comments, follows.

Reviewer #1:

The authors have addressed most of my concerns, but not all, and some new concerns are raised below. I appreciate the work done, but here I will focus on the parts that need attention before publication.

Many thanks for the new comments and we apologize for skipping any of your early suggestions.

Manuscript Figure 6. If the Ru-alkene pi complex is Ru(II), then the pi allyl hydride complex is Ru(IV). I asked that this be corrected but it was not. Moreover, the arrows that are trying to show atom movement should be deleted entirely; I apologize for not being clearer when I wrote that they should be corrected. Such arrows are supposed to show electron movement, and the present arrows do not, nor is it really feasible to show it.

Thanks for the comments. The Reviewer is right, and the arrows should be erased. Following the Reviewer's suggestion, we have done that. Regarding the Ru oxidation state, we have not any evidence yet that the process goes through Ru(IV) but, as the Reviewer points out, Ru(II) could not be the Ru oxidation state throughout the entire catalytic cycle, thus we have deleted the Ru oxidation state in Figure 6c (now just Figure 6) and we have indicated in the caption the possibility but not certainty that Ru(II) is the catalytic active species.

Figure S11. The expected masses are given with enough significant figures, and most of the experimental masses are visible, but not for the peak 324. Most importantly, the "envelope" of all the peaks due to Ru isotopes (mostly) are still not visible. Please make enlargements so one can see all the peaks.

Thanks again for the observation. Following the Reviewer's suggestion, we have included an inset in Figure S11 (new Figure S13) with the amplification of the peak at 324.1402, where the "envelope" due to the Ru isotopic distribution can be seen.

Figure S15 and added text in manuscript, experimental section, pp 23-24. The explanation alone is really not sufficient. Are the authors meaning the 30 to 60 min portion of the plot? If that is the case, it should be made clear.

Sorry for not specifying the kinetic points from which initial rates were calculated. Following the Reviewer's suggestion, we have now included the exact time intervals in the manuscript, experimental section, pp 23-24: "(0-10 min for 6, 5-20 min for 6-d²)."

Manuscript page 17. Rather than writing d16 or d14, one usually writes 16-electron or 14-electron – please change. The d superscript notation is used to describe the electron count of the metal, not the whole complex.

The Reviewer is absolutely right, sorry for these mistakes. The changes have been made.

Figure S18. I thank the authors for including some DFT calculations, but definitely more needs to be done before publication. Please see points 1 to 3.

Thanks for the insightful comments.

1. why Ru(I)? There is no explanation why this oxidation state was selected.

Ru(I) was originally selected since we are not totally sure of the catalytically active Ru oxidation state but we know is Ru(II) or lower. However, Ru(II) is a much more stable oxidation state and the most plausible for the reaction. We have now included the corresponding calculations for Ru(II) in Fig. S19, and we have left the previous results for Ru(I) for the sake of comparison.

2. the calculations seem to show a "ligandless" Ru atom interacting with 1-butene, but what I want the authors to calculate is what they are actually proposing on the basis of experimental evidence such as mass spectrometry, for example with four or three bound alkenes, giving 16-electron or 14-electron species that are discussed in the text.

Sorry for our misunderstanding in the previous revision. Following the Reviewer's comments, we have now included the DFT results with three and four alkene ligands, and the results are shown in the new Figures S19. It can be seen there that the Ru(II) complexes with 3 or 4 alkenes suffer a smooth complex formation and 1,3-hydrogen shift steps (not higher than 4 kcal·mol⁻¹) and then a barrierless isomerization. Selected Ru-C and Ru-H bond distances have been included in the Figure.

3. Customary presentation of computational results includes the following:

3a. Relevant details of the computational method, including basis sets

These computational details are now included in the methods section of the manuscript. Our apologies for not including these data in the first revision of the manuscript.

MINISTERIO
DE ECONOMÍA
Y COMPETITIVIDAD

UNIVERSITAT
POLITÈCNICA
DE VALÈNCIA

INSTITUTO DE
TECNOLOGÍA
QUÍMICA

3b. Tables of atomic coordinates and E and G information for each.

The atomic coordinates are included as the new Table S3 in the Supplementary Material.

3c. It would be nice to include some key bond length information.

Selected bond distances for Ru-C and Ru-H have been included in Fig. S19. The numeric values are:

Ru-C in 3alkenes-Rull 2.22 Å; Ru-H in 3alkenes-Rull 1.54 Å

Ru-C in 4alkenes-Rull 2.57 Å; Ru-H in 4alkenes-Rull 1.97 Å.

Minor typos: Fig. 2 caption line 5, “the Ru valence state” should be “the Ru oxidation state” .
Similar comment on next page, 4 lines from bottom. “low-valent” not “low-valence” . Last line of the paragraph, “unprecedented”.

Many thanks for the observations. The changes have been made as suggested.

Reviewer #2:

cPer the editor’s request, I have taken a look at the revised manuscript “*Parts-per-million of ruthenium catalyze the selective chain-walking reaction of terminal alkenes*”, at the authors’ thorough reply to my and referee #1’s first round of comments, and at the Supporting Information. I find that the authors have carefully addressed my, and is would also appear much of referee #1’s, comments and suggestions. I conclude that this paper is close to publication, following just a small number of additional suggestions, provided below.

Many thanks for the nice and positive comments.

Additional Suggestions

(1) P. 13 in the Bold Heading and Figure 6 c: insert “proposed” to say “Proposed Reaction Mechanism”. No mechanism is exactly correct to start, so that phrasing it this way will avoid criticisms later on when the mechanism get updated by additional studies.

Thanks for the suggestion, we have made the change.

(2) Most of p. 14, 15, and the ½ of p. 16 up to the “The mechanism of the isomerization reaction was studied....” is probably best placed in the SI, as it only indirectly bears on the mechanism. It would be better to just start the Reaction Mechanism discussion on p. 13 at the above sentence

("The mechanism of the isomerization reaction was studied..."), and then put the CV and the IR studies in Figure 6 into a separate, later, brief figure with much of the details in the SI. The revised Figure 6 can be just the Proposed Reaction Mechanism". These changes will tighten the manuscript significantly from p. 13-16, and by doing so not have the mechanistic discussion detract from all the nice work presented p. 1-13 which is the main part of the paper.

Thanks for the comments. Following the Reviewer's recommendations, we have changed all this part of the manuscript, placing the text in pages 14-16 into the SI (as comments in the corresponding Figure) and leaving Figure 6 with just the proposed reaction mechanism, placing the CV and FTIR results as the new Figures S14 and S15 in the SI. We have properly re-numbered all the rest of Figures in the SI.

(3) P. 17, bottom: the authors should find some references and cite them (and check) the Ru-H and Ru-olefin BDEs that I quoted to them from memory and as ballpark numbers. These values are not particularly easy to find as they are spread out in the literature. But, look for them, starting with J. L. Beauchamp's *Chemical Reviews*, 1990, 90, 629-688, and papers, again just to start.

a. In Table 13 therein he lists in Ru-H = 234 ± 21 kJ/mol (so 60 ± 5 kcal/mol) and Ru(II)-H+ as 172 ± 13 kJ/mol (so 41 ± 3 kcal/mol)—note these are other-ligand-free binuclear species in the gas phase.

b. As for Ru(II)-Olefin BDEs, one place to start is Nechaev, Mikhail S. et al. *J Phys. Chem* **2004**, *108*, 3134-3142.

c. Again, be sure to search the rest of the literature for these Ru-ligand bond energies.

Many thanks for the comments. Please notice that we have a reference number limitation in the journal and we already had to remove some references in the first revision to make room for the new references proposed by the Reviewer. We have checked that the numbers given in those references for Ru(II)-H (as the Reviewer kindly provided, ~ 70 kcal/mol) and Ru-H [41.7 kcal/mol for ethylene bound to Ru(CO)₄, in the Nechaev, Mikhail S. et al. paper] are in line with those commented before, so it is fair to leave them as a guidance number. We have included the L. Beauchamp's *Chemical Reviews* paper as the new reference 54 in order to direct to the Readers if they would like to check the numbers. In any case, these energies are not the aim of this work and, as mentioned by the Reviewer, the proposed reaction mechanism cannot eclipse the main experimental part of results, thus please allow us not to enter in more details about this part.

(4) P. 4-5, I still think that Figure S3 in the Supporting Information should be moved to the main text, around p. 4-5, as seeing those structures there, then learning that all the complexes behave similarly under catalytic conditions at 150-200 oC, is a very powerful, telling part of the paper.

Thanks again for the comment. Following the Reviewer's suggestion, we have placed Figure S3 in the main text, as a part of Figure 2 (page 4).

In summary, with attention to the above points and any additional comments from referee #1, I am happy to see the final, revised paper published in *Nature Communications*

REVIEWER COMMENTS

Reviewer #1 (Remarks to the Author):

The revisions are very helpful.

I have fewer comments this time.

Fig. 6 – there needs to be some revision. Count carbons carefully. The starting terminal alkene has $m+3$ carbons in the chain attached to R. The first pi allyl H intermediate structure is fine, but the complex of the once-isomerized alkene now has only 4 carbons in the chain, and the same is true for the next alkene complex, and for the free alkene. The intermediates after the pi allyl H one do not have the () m anymore....

Page 14 line 16 ff sentence beginning with “In accordance with this....” I do not understand what is meant by a high excess of alkene being needed to recover catalytic activity. I think I would delete the sentence altogether because transition to the next one (“Figure 6 shows the plausible mechanism...”) will then be seamless.

Later in this paragraphline 5 from bottom of the page “until findswhere definitely stops” is missing a subject in two places, perhaps “until the catalyst findswhere the catalyst definitely stops” will work.

Fig. S13 – thank you for the inset now showing the envelope for 324.1402.....there should be insets also for the other three circled low-intensity masses that are assigned to Ru(1)4-H, Ru(1)2-H and Ru(1)3-2H. I mistakenly focused on 324 only in my previous comments, for which I apologize.

Fig. S19 – thank you for the added calculations. It is interesting that on the complexes, in the more crowded tetrakis alkene case the isomerized complex energy is increased to a greater extent. Could you make the structures bigger for better visibility? I have a big monitor and the file at >200% and it is still a little hard to see the details.

It would have been better to calculate the methyl eugenol-derived species rather than the 1-butene ones, because the mass spectrometric evidence is not for butene complexes. The added steric bulk of the aryl group will certainly change things a lot! Now that the authors have the butene-derived structures, it should not be difficult to calculate the real thing, maybe just with phenyl instead of with the two methoxy groups, but the methoxys should not add much computing time.

Minor changes

Page 6 first line – “(for structures see Fig. S3)” should be deleted

Page 13 line 2 under Fig. 6 “voltammetry” should be changed to “volammetric” to match the other adjectives in the list modifying the noun “experiments.”

MINISTERIO
DE ECONOMÍA
Y COMPETITIVIDAD

UNIVERSITAT
POLITÀCNICA
DE VALÈNCIA

INSTITUTO DE
TECNOLOGÍA
QUÍMICA

Dr. Antonio Leyva Pérez
Tenured scientist / Distinguished Researcher
Instituto de Tecnología Química UPV-CSIC
Avda. de los Naranjos s/n,
Universidad Politècnica de Valencia
46022 VALENCIA, SPAIN

Let us thank again to the Reviewer for the comments. A point-by-point answer follows.

Reviewer #1 (Remarks to the Author):

The revisions are very helpful.

I have fewer comments this time.

Fig. 6 – there needs to be some revision. Count carbons carefully. The starting terminal alkene has $m+3$ carbons in the chain attached to R. The first pi allyl H intermediate structure is fine, but the complex of the once-isomerized alkene now has only 4 carbons in the chain, and the same is true for the next alkene complex, and for the free alkene. The intermediates after the pi allyl H one do not have the () m anymore....

The Reviewer is right, sorry for this mistake. We have carried out the changes, and now the carbon count should be right, as well as the significance of “m” during the whole cycle. We have also changed the Figure caption, accordingly.

Page 14 line 16 ff sentence beginning with “In accordance with this....” I do not understand what is meant by a high excess of alkene being needed to recover catalytic activity. I think I would delete the sentence altogether because transition to the next one (“Figure 6 shows the plausible mechanism...”) will then be seamless.

Thanks for the comment. Following the Reviewer’s suggestion, we have removed the sentence from the manuscript.

Later in this paragraphline 5 from bottom of the page “until findswhere definitely stops” is missing a subject in two places, perhaps “until the catalyst findswhere the catalyst definitely stops” will work.

Thanks again for the corrections. The suggestion has been included in the text.

Fig. S13 – thank you for the inset now showing the envelope for 324.1402.....there should be insets also for the other three circled low-intensity masses that are assigned to Ru(1)4-H, Ru(1)2-H and Ru(1)3-2H. I mistakenly focused on 324 only in my previous comments, for which I apologize.

The Reviewer is right in having all peaks amplified, and we have changed the Figure accordingly. Fig. S13 shows now the four peaks with the corresponding amplifications, as insets.

Fig. S19 – thank you for the added calculations. It is interesting that on the complexes, in the more crowded tetrakis alkene case the isomerized complex energy is increased to a greater extent. Could you make the structures bigger for better visibility? I have a big monitor and the file at >200% and it is still a little hard to see the details.

The Figure has now been made bigger for a better visualization, twice the original size.

It would have been better to calculate the methyl eugenol-derived species rather than the 1-butene ones, because the mass spectrometric evidence is not for butene complexes. The added steric bulk of the aryl group will certainly change things a lot! Now that the authors have the butene-derived structures, it should not be difficult to calculate the real thing, maybe just with phenyl instead of with the two methoxy groups, but the methoxys should not add much computing time.

It is a common practice in chemical research to carry out the DFT calculations with the smaller reactant in order to save computing resources. We think that the choice of 1-butene is correct. Indeed, the calculations with 1-butene took nearly three months and are perfectly valid to explore the mechanism, since simple linear alkenes are very reactive under typical reactions conditions (see Figure 3, compound 21). However, in order to be completely sure that 1-butene is a right reactant to study, we have performed the reaction with this alkene, and the results are positive. We have included a new sentence in the manuscript to note that 1-butene is also reactive: “this alkene is also reactive under the conditions in Fig. 4A, using Ru/C as a catalyst”.

Following the Reviewer’s suggestion, we have considered to compute methyl eugenol. However, this substrate is three times heavier than 1-butene, and considering that four molecules are present in the different intermediates, we are increasing the computational calculation at a level that will take several months to have any result. We think that the work already made with 1-butene is enough to support our experimental work, which is the main part of the study.

Minor changes

Page 6 first line – “(for structures see Fig. S3)” should be deleted

Many thanks, the change has been made.

Page 13 line 2 under Fig. 6 “voltammetry” should be changed to “volammetric” to match the other adjectives in the list modifying the noun “experiments.”

Thanks again, change made.